# The Tree of Abundance: On the Indigenous Emergence in Contemporary Latin American Art

Miguel L. Rojas-Sotelo

Duke Center for Latin American and Caribbean Studies, Duke University, Durham, NC 27708, USA; mlr34@duke.edu

**Abstract:** The Tree of Abundance is an origin story for many nations in the Amazon basin. It recounts a time when all people(s) lived under a mother tree, until those with an ax arrived and the tree collapsed. This is the act of coloniality, which produced a new landscape. The story serves as a conceptual metaphor to analyze the production of an emerging generation of contemporary visual makers of indigenous origin. These cultural producers are set in a historical context, which represents long temporalities of cultural-production resistance and re-existence in Latin America (called here Abya Yala). The text introduces a way to rethink contemporary art in the region under conditions of coloniality and names the artists "embodied territories" since they have particular connections to the places they live and work. This article is organized into three parts presenting artwork by several indigenous and intercultural subjects (with emphasis on those living in indigenous territories of Colombia): (1) A short genealogy from modernity to contemporaneity brings indigenous cultural production to the academic space as another source for a critical understanding of the lived experience in Abya Yala. (2) An account of themes derived from the contested histories highlights how indigenous and intercultural artists produce responses to them. (3) The genealogy and themes are then set in spatial terms offering two case studies, on one hand, the toppling of historical figures by indigenous activists as performance in the public space and, on the other, the exhibitions "Visual Sovereignty" and the "Indigenous Salon Manuel Quintín Lame". The article concludes stressing how this emerging generation builds on long genealogies of sovereign representation, responding with a wide range of contemporary means (visual, textual, bodily, and multimedia) to issues that still affect their communities (land grabs, resource extraction, racialization, marginality, etc.). Adaptation, resistance, and re-existence occur when embodied territories recognize historical realities (time), location (space), and forms of liberation (action) within coloniality.

**Keywords:** indigenous studies; indigenous art; art history; interculturality; Latin American art; Colombian contemporary art; resistance; contemporary art; decoloniality; epistemic reconstitutions

## 1. Introduction

How do we account for the emergence of indigenous visual producers within the time frame of contemporary art in Latin America? The goal of this text is not to identify the recent emergence as a nuance but as a continuum of the will of indigenous subjects to represent their own experience within the conditions of coloniality and to return sovereignty of representation to them. The concepts of art, contemporary art, art history, and the artist must be revisited, or at least called into question, since they are the result of the forces of the constitution of the West and enforced via coloniality upon the indigenous peoples of the Americas. On the one hand, for many indigenous languages, a direct translation of the term "art" does not exist, less one that corresponds to a historical categorization such as contemporary art. Both respond to teleological ways of telling history in contradiction to the cyclical temporalities of most indigenous peoples. On the other, the modern notion of the artist is based on the idea of authorship and the "male genius", which do not adequately define cultural production in indigenous communities.[1] The short span of this article cannot

develop on these issues which have already been confronted by scholars of the decolonial aesthetics option.[2] However, this article would present, in context, some of the flows that have allowed the presence and participation of contemporary visual producers of indigenous origin in Abya Yala (Latin America), with special attention to some located in territories of the Andes and the Amazon.[3]

Contemporary art is usually dated to have begun at the end of War World II (mid-twentieth century), signaled by the shift of the location of cultural centers (from Europe to the United States), by abstract expressionism, pop, and minimal art. In other accounts, contemporary art is even equated with the fall of the Soviet Union (1989–1991), the consolidation of the era of "late capitalism" (neoliberalism), the so-called end of history, and the network society. More recently the concept has been amplified to say that contemporary art is art made today by living artists (Walker Center); or contemporary art means "the art of today", including artwork produced during the late 20th and early 21st centuries (IESA International).[4] Generally, contemporary art is defined as art that responds to the times we live in, acting on broad contextual and conceptual frameworks and artistic practices. As for our discussion about Latin American contemporary art and the emergence of indigenous contemporary art, I will use these definitions to expand and contrast the timelines as needed.[5]

Apart from the success of Aboriginal art in Australia, which started in the late 1970s as an apology for their cultural destruction under British colonization, and the later commercial success of some of its practitioners in the art market (starting in 2007), indigenous contemporary art is an outlier. It is relevant to mention the discussion that took place during MOMA's 1984 exhibition "Primitivism in 20th Century Art: Affinity of the Tribal and the Modern", as well as those around the exhibitions "Magiciens de la Terre" (1989) and the 1991 Havana Biennale organized under the label "desafío a la colonización" (Rojas-Sotelo 2009).[6] These global art events brought for the first time to art audiences and cultural critics, if briefly and with limitations, the work of indigenous artists (or magicians), their practices, and realities. Today, new conditions have allowed a generation of indigenous creators (embodied territories) to emerge, positioning themselves in multiple spaces within and outside the art world.

### 1.1. Abya Yala

It is clear that indigenous peoples in the Americas were conquered, repressed, silenced, reduced, integrated, and/or marginalized (even after independence) by colonial (and internal colonial) powers. For many, they are an issue of the historical past. However, it is also a fact that indigenous peoples have survived coloniality and have developed forms of resistance, subsistence, and re-existence at the margins of and even within modern societies. For instance, indigenous intellectuals in the mid-1980s preparing for the 500-year encounter/clash anniversary came together to rename the continent (2017).[7] The name "America" had obliterated their histories and ways of life. Located at the Darién Gap, the belly button of the continent, between what is today Panamá and Colombia, are the Gunadule (Kuna) peoples, who call their land *Abya Yala*—a linguistic expression that literally means "land that bleeds". As Rocha Vivas (2018) points out, at present, the term has assumed different meanings according to the ecocultural contexts of the continent. The Aymara leader Takir Mamani (Bolivia) defines Abya Yala as a "land in full maturity". On the other hand, Armando Muyolema from the Kichwa (Ecuador) suggests an interpretation of Abya Yala as an epistemological and social form, an alternative of Western civilization. Finally, in the linguistic reading proposed by the Gunadule scholar Mani-binig-diginya (Abadio Green), Abya Yala carries in its etymological entrails the meaning of "birth, blood–bone of the earth" and, at the same time, "the collective loss of blood after confrontations because of the European invasion" (2018: 57). Hence the term is capable of absorbing the violence of the conquest and independence, and the cultural resistances across the continent. This epistemological, as well as political, renaming of Latin America as Abya Yala will be maintained across the text.[8]

On 1 January 1994, the Ejército Zapatista de Liberación Nacional (EZLN) declared war against the Mexican State, the same day as the North American Free Trade Agreement (NAFTA) was implemented between the U.S., Canada, and Mexico. In addition to the economic, social, and cultural implications the declaration unfolded, it also allowed the social and cultural realities of the South to be visible.[9] Following the incident, a cultural emergence took place, embodied in the form of orality (radio), visual (murals), and audio-visual production, and the establishment of networks that brought the Mayan struggles to other indigenous peoples of the South, such as those already taking place in the Colombian Andes (Rojas-Sotelo 2020). From then on, an emergence of cultural production about, with, on, and from indigenous peoples of Abya Yala coalesced, connecting lines of cultural resistance across the territories of the continent.

Contemporary Brazil and Colombia contains 254 indigenous nations and 104 indigenous nations recognized in each country, respectively, albeit the indigenous population represents only 0.4% and 3.43% of the total population in each nation. These communities continue to suffer disproportionately from internal political, social, economic, and environmental conflicts. In Brazil, more than 50% of indigenous peoples live outside their ancestral territories; in Colombia, between 1993 and 2020, the forced displacement of indigenous communities grew from 7.42% to 21.42%, thus adding to the loss of territory, the destruction of habitats, and therefore the loss of forms of cultural and territorial sovereignty. Nonetheless, in Brazil and Colombia, thanks to new political constitutions written in 1988 and 1991, indigenous peoples were recognized as political actors, and other countries followed.[10] This fact also made visible the struggles of indigenous communities amidst the escalation of violence in part due to the expansion of the economic frontier. Extractive operations (coal, oil, gas, gold mining, timber, cattle ranching, mono-crops) as well as a service economy, both market-based approaches, rule the economic (and cultural) discourse.

The 2007 Declaration on the Rights of Indigenous Peoples proclaimed by the United Nations (United Nations 2007) stated that "indigenous peoples have the right to practice and revitalize their cultural traditions and customs. This includes the right to maintain, protect, and develop the past, present, and future manifestations of their cultures, such as archaeological and historical sites, artifacts, designs, ceremonies, technologies, and visual and performing arts and literature". In 2016, the Organization of American States (OAS 2015) pronounced the "American Declaration on the Rights of Indigenous Peoples". With a basic framework, institutions and policies have been developed, and communities, agents, and cultural producers have been taking those spaces by storm, making visible their realities, demands, and also their cultural production.

### 1.2. The Tree of Abundance

The Tree of Abundance, or *Monilla Amena*, is a Uitoto myth (with manifestations in the Amazonian communities Andoque, Bora, and Muinane, among others) that addresses the origin of the Amazon and its people. "The story tells that if one rises in the territory, one sees a reality in the form of a tree" (Urbina Rangel 2010).

The story refers to the original tree that fed all creatures until it was taken down by a conflict—the act of coloniality. Once it hit the ground, it created three rivers; the one on the surface is the Amazon (the one above is the climate system, and the one undergrown is the realm of spirits). This myth originates the idea of Pan Amazonia, a great organism, a macrosystem, that is comprised of an integrated set of ecosystems.[11] The Tree of Abundance is used here not as a methodology but as a metaphor (allegory if we must) of the richness and autonomy of cultural expressions within the multiple indigenous nations in Abya Yala. The myth behind the tree has survived as a story of origin, passed by oral means and incarnated via ritual across generations; it announces an alternative ontology. The myth is shared orally, in the maloca (the ceremonial house in the Amazon) after the intake of sacred plants. It is similar to the way other origin myths are also shared (in the Judeo-Christian biblical narrative, a burning bush spoke to Moses, revealing Yahweh's name, commanding him to lead the Israelites out of Egypt and into Canaan).

Contemporary cultural producers of indigenous descent have represented the myth in visual form. In *The Tree of Life and Abundance* (2012), Mogaje Guihu (Nonuya and Muinane) presents the centrality of the myth for his people (Figure 1). The tree was located at the center of the territory and provided food and shelter to all living beings—but not all (the ones on the other side, the ones that used axes). Each branch produced different food and belonged to a particular people—including nonhuman ones. The painting illustrates a set of instructions for an organized society, where all are related and governed by those beyond human entities (spirits). When hunger came to the people of the ax and because they did not understand the calls from the spirits, they cut the tree, ending the abundance. The work illustrates the instructions expressed in the myth, instructions that allow man not only to cohabit with other humans but with the entire natural world. Mogaje Guihu's (1943) work derives from natural illustration and although it is documentary in nature points to another ontology, one that parallels the Western one. His botanical knowledge is shared via illustrations that are situated and contextualized. Guihu was born in the Amazon as a member of the Nonuya people and lived in the forest for six decades before arriving in the city as a forced displaced subject. Guihu is an organic botanist and healer who through his visual work tries, in an exercise of translation, to present other forms of knowledge: " . . . to be a traditional doctor, to know many kinds of medicinal or nonmedicinal plants, to build the maloca, is to be able to share, teach, and earn, both in the body-healing and mind-education. That place (the maloca in the middle of the forest) for us means what the school, the university, is for you; for us, that is the maloca" (my conversation with Guihu 2017).

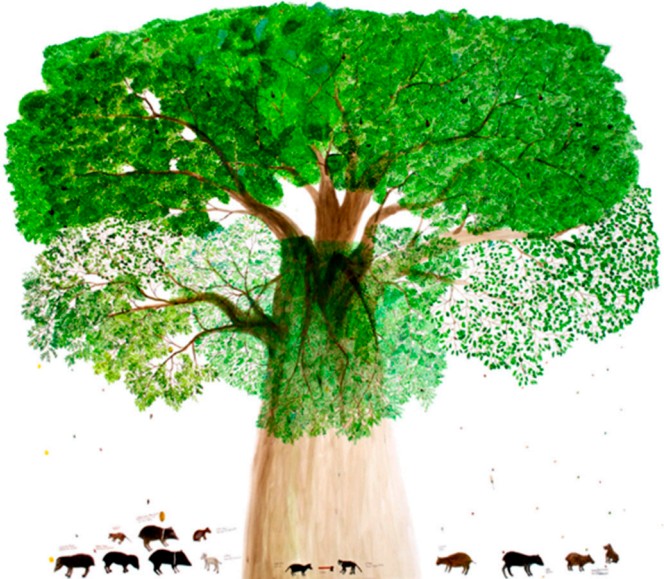

**Figure 1.** Mogaje Guihu (Abel Rodríguez Muinane), Colombia. *The Tree of Life and Abundance*. Ink on paper. 200 × 150 mts. 2012. Collection Tropenbos. Photo: Tropenbos.

In his *Monilla Amena* (2016), Brus Rubio Churay illustrates a tree made of children, almost in the baroque tradition of cherubs, representing all the peoples of the Amazon, their past and future (Figure 2). The tree emerges from a hole (a vaginal form) from which all is created; there is a shaman and a group of rodents that in versions of the myth are the culprits of its falling. The tree's roots are made of aquatic beings (when the tree falls, its branches and leaves become rivers and fish). Rubio Churay's work is allegoric and cosmologic; he states that it "reflects a great cosmic happiness because it is inspired by gods and important mythic people, by festivals and rituals, by the tasks of farming and minga (cooperative work), by the magic and beauty of fish and animals, by the song, vision, and sacred work of my ancestors" (Rubio Churay 2017). In this way, his work is as contemporary as it is ancestral, bridging two cosmovisions that give subjectivity to other ways to see and be in



the world. Rubio Churay (1984) is a visual artist of the Muriu-Bora and Uitoto peoples from Loreto in Peru's northern Amazon (exiles of the rubber genocide in la Chorrera, Colombia). Rubio Churay lives and works between Paucarquillo (his hometown) and Lima, where he migrated in 2009. Between the city and the jungle, he brings the world of the modern man together with the myths and goods from the realm of the forest.

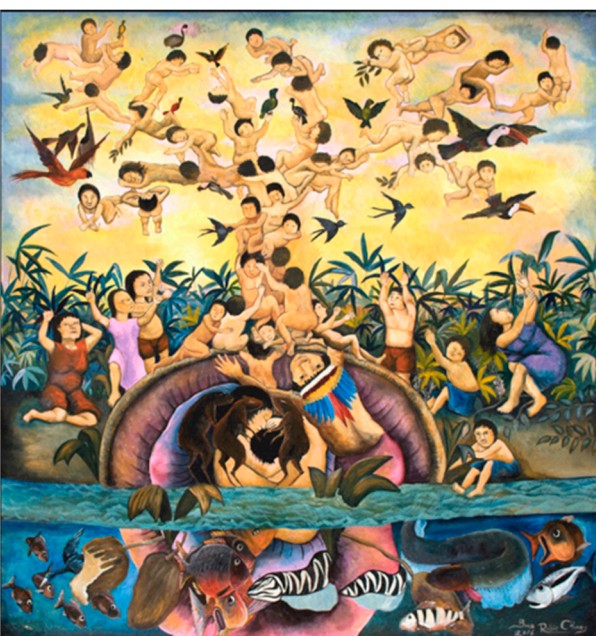

**Figure 2.** Brus Rubio Churay (Bora-Uitoto). *Monilla Amena. Árbol de la abundancia*. 100 × 105. Acrylic on canvas. 2016. Private Collection. Photo: Brus Rubio Churay.

The Tree of Abundance is then a manifestation of a particular territory that was transformed by a violent act and that today is shared by multiple nation-states (and claimed by the world as a key space for biodiversity and climate mitigation). As a matter of fact, another translation for Abya Yala is "living land" or "land that flourishes", an expression of a claim to sovereignty.

What follows is a short genealogy of indigenous cultural production in Colombia divided into three parts. Although a linear history, the genealogy tries to bring indigenous cultural production from modernity to contemporaneity to the academic space as another source for a critical understanding of the lived experience in Abya Yala. Here, subjects are understood as cultural producers, expanding the definition of the artist since they work in multiple registers: the scriptural, the visual, the audiovisual, the intellectual, the bodily, the oral, and even the political. The exercise places indigenous producers into a dialogue calling them embodied territories.[12]

This genealogy is followed by a brief account of themes derived from the contested histories still affecting embodied territories. This section underlines some of the connections between issues (historical, social, political, cultural) and cultural producers, and how they manifest such relations between history and their bodily and intellectual practice in multiple mediums. Finally, the genealogy and themes are set in spatial terms as these embodied territories are present in both the public space and the art world. This section offers, in brief, how contemporary producers are interacting with both spaces and discusses a couple of examples of how this emergence is taking place: by toppling statues of colonizers in the public space and organizing art exhibitions such as Visual Sovereignty and the Indigenous Salon Manuel Quintín Lame. In that way, a multilevel archive is being constructed with the participation of cultural subjects that move freely between cultural worlds and political borders.

Contemporary indigenous visual and audiovisual production is beginning to be recognized. Although incomplete, this text tries to collect and present in historical terms the production of creators connected to long-term processes of cultural visibility in their territories (with a focus on some territories of Colombia). In a narrative way, and in a linear historical key, relevant moments of this emergence are underlined, which also allows new approaches and research to materialize. In short, indigenous artistic production is a tree of abundance in search of cultural sovereignty.

## 2. Fabulations, Listening, and Intercultural Dialogue: A Short Genealogy

Fabulation is any story that challenges its two main assumptions, form and subject matter, allowing other worlds to be seen and told (Clute 2022). It is for instance a way to document reality that takes place at the edge of conventional genres, outside of Western modernity. Gilles Deleuze defines fabulation as the "becoming of the real character when he himself starts to 'make fiction', when he enters 'the flagrant offense of making up legends' and so contributes to the invention of his people" (Hongisto 2015, p. 67). This is in part what contemporary indigenous visual producers are making, revealing worlds and inventing people. Vilma Almendra, the Nasa-Misak intellectual from Cauca, and Ariruma Kowii, the Kichwa intellectual and poet from Ecuador, remind us how listening is one of the epistemological strategies of the South (since myths are not read but heard to be embodied) (Ferrari 2020). It holds because when mother nature speaks, you just hear and then act in accordance. In addition, the relational ontologies (in terms of Arturo Escobar (Escobar 2012)) of indigenous peoples are based on intercultural dialogues, which make clear that there is a logic for understanding the world and relating to nature that does not correspond to a capitalist one (2012).[13] According to Afro-Colombian intellectual Adolfo Albán Achinte (Albán Achinte and Rosero 2016), an alternative to development and progress can be based on "interculturalizing" our projects, so that they do not threaten life. If processes of historical reparations to ethnic groups are met, to overcome the structural inequities that deny their existence and enable respect for nature, beyond the abuses of extractivist megaprojects, we can start to build a new world, to re-exist (2016: 33).

Before jumping into the contemporary moment, this genealogy begins with the argument that a desire for representation and documentation (in an oraliterographic form) occurred with the mural practices, goldsmithing, stone carvings, and the production of the material culture of the original peoples of the land. If so, the spectrum of cultural agents (artists), times, and spaces grows exponentially. In the last decade, the re-encounter with the majestic mural paintings in the Chiribiquete and La Lindosa mountain ranges in the Colombian Amazon (states of Guaviare and Caquetá) makes clear that this will of representation and documentation went well beyond ritualistic practices (Figure 3). They signal a type of political aesthetic (documentary and symbolic) that has been present in the territory for possibly tens of thousands of years. A palimpsestic practice presents iconographic, archetypical, and archeological challenges due to representation and location.[14] In this genealogy, the murals are at the beginning of the timeline (around twelve thousand years ago according to current science).

It is worth clarifying that the concept of art in its modern form is a European construct of the eighteen century (Kantian aesthetics).[15] However, its notion arrived with the European conquest (found in several chronicles of the Indies and derived from its latin root 'ars', meaning skill, craft, art, coined in 13th century European manuscripts) and was consolidated during colonial times and confirmed in the republican era. It is also important to clarify that even today for many indigenous communities, there is no translation of such a concept (art) in their languages.[16]

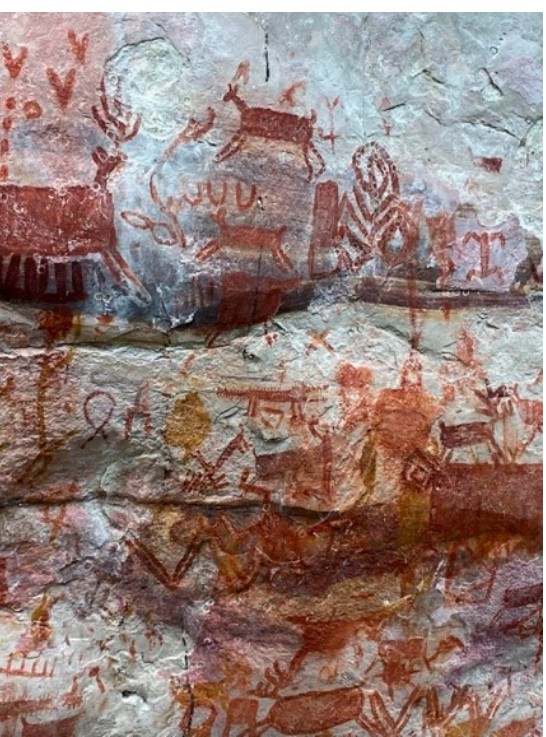

**Figure 3.** Murals in Serranía de la Lindosa. Sabana del Venado. Photo: Sara Sofía Rojas. 2022.

Across the region and until the republican era, the teaching of art took place in apprenticeship studios. In modern Colombia, the creation of the National School of Fine Arts in 1885 changed the teaching to a derivative academic practice. Its promoter and founder, the artist, caricaturist, writer (and commander of the Army) General Alberto Urdaneta (1887), on the inauguration of the first visual art exhibition in the new republic in 1886, said

> When the great Spanish civilization hit its targets in the green forests of America, here art had no more samples than the few advances achieved by the Chibcha nation: its primitive ornamentation, its scrolls, and fretwork, its crude suns of gold and their imperfect clay idols. There is, near this enclosure, the first painting, the first canvas painting, which replaced the red drawings of the children of Chiminigagua: the banner of (Gonzalo Jiménez de) Quesada: the Christ of "the conquest". From this canvas dates, it can be said, the history of art among us. *Papel Periódico Ilustrado*.[17]

This event left the visual production from and about indigenous peoples in limbo for decades. However, in 1922, the engineer and historian (the pioneer of archeology in Colombia) Miguel Triana Cote published his book *La civilización Chibcha* (*The Chibcha civilization*), which collected forty years of fieldwork and interest in the pictograms and petroglyphs located in the Andean highlands, Boyacá and Cundinamarca, continuing the line of research of Ezequiel Uricoechea Rodríguez in the 19th century. It supported the international success of sculptor Rómulo Rozo (of indigenous origin), who with his *Bachué* (1925) won the Seville World Fair prize of 1929 (Figure 4). Finally, recognition and interest in indigenous themes and aesthetics entered the national discourse, and an entire generation followed, known as the Bachué generation (an indigenista movement). The arrival of Argentinean art historian and critic Marta Traba to Colombia in 1954, with an interest in international modernism, would reverse the tendency and promote a generation of artists related to such flows, condemning the Bachué generation to oblivion. Similar histories of force modernization and the silencing of indigenous emergencies happened in other territories of Abya Yala by the second half of the 20th century.[18]

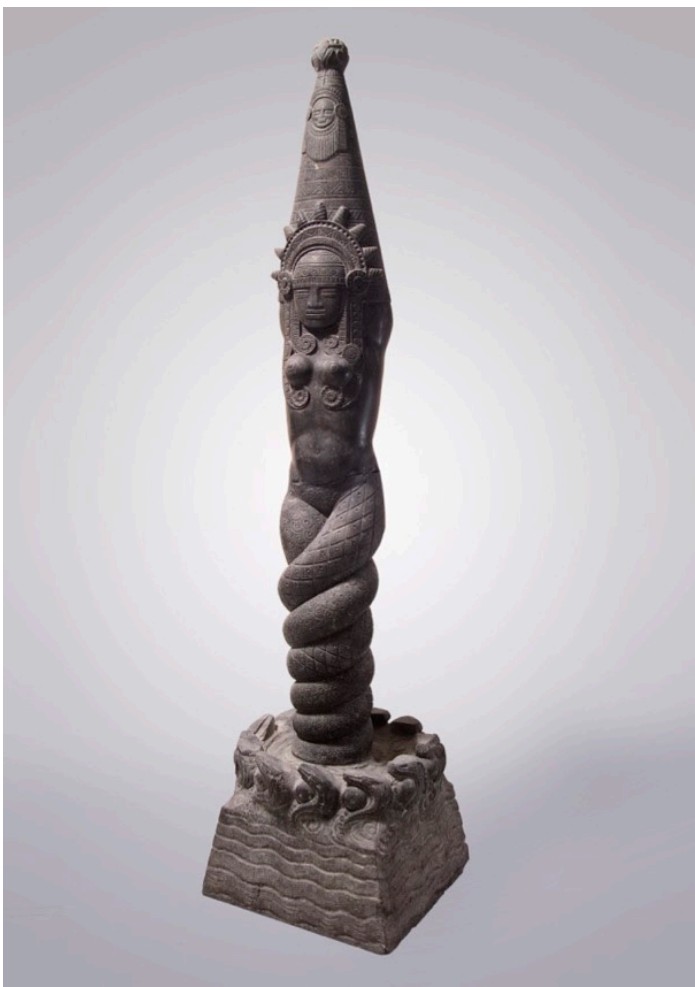

**Figure 4.** Rómulo Rozo. *Bachué: Diosa generatriz de los Chibchas*, 1925. Granite carving. Private collection. Photo: Dantelectríco (Creative Commons).

In 1960, the exhibition entitled "3000 Years of Colombian Art" was shown at the Lowe Gallery of the University of Miami. Produced by a group of researchers (among others, archaeologists Hernández de Alba, Reichel-Dolmatof, Duque Gómez, and Jiménez Muñoz), it presented Colombian art from the first evidence of the Augustinian civilization (4 century BC) to the generation of artists represented by Marta Traba (the exhibition was partly financed by the North American oil company in Colombia, ESSO, and intended to introduce this new generation backed by Traba). In response to it, José Gómez Sicre organized the exhibition "500 Years of Colombian Art" at the Pan American Union in Washington, D.C. in 1962, which erased the indigenous (preconquest) dimension of the history of art. The internationalization and modernization of Colombian art occupied the following decades. Biennials, triennials, and salons would seek to affirm a Western, capitalist, and modern industrial culture (represented in the Coltejer Biennial in Medellín, the Cali Graphic Triennial, the National Salon, the Bogotá Biennial, and the establishment of the Bogotá Museum of Modern Art (MAMBO), in addition to several museums of modern art in different regions of the country).

It is during the 1980s, in preparation for commemorating 500 years of resistance, the anniversary of the encounter/clash, indigenous and intercultural intellectuals, writers, and cultural producers began working on open agendas, cultural and political, for the recovering of the means of representation. During the 1970s and 1980s, many indigenous territories in Colombia suffered an increase in violence due to the encroachment upon lands and the expansion of the economic frontier (drug trafficking and guerrilla war), which pushed for organization and reaction (as in the case of the Organización Nacional



Indígena de Colombia (ONIC), the Consejo Regional Indígena del Cauca (CRIC), and the Indigenous Guard, among others). The work of philosopher and ethnographer Fernando Urbina Rangel (on collecting stories, documenting petroglyphs, rock art, indigenous rituals, and traditions via photography, oral history, and performance) at that time consolidated a new generation of situated visual ethnographers and art historians. It followed a wave of political and administrative decentralization in the arts with the creation of regional art salons, organized by Colcultura (National Institute of Culture), starting in 1986. At the same time, the work of visual producers such as Antonio Caro Lopera brought recognition to the ongoing demands of many communities. Caro Lopera's (1950–2021) *Proyecto Maíz*, 1981–2021, and *Proyecto 500*, 1988–1992, established a critical space for indigenous representation. His *Homenaje a Manuel Quintín Lame* (*Homage to Manuel Quintín Lame*) in 1992 brought to the center echoes of cultural production and the resistance of the indigenous peoples of the Andes.[19] In addition, Caro Lopera developed collective workshops with, in, and about the indigenous communities of Putumayo and Cauca. With the signing of the 1991 Constitution, a program called "Crea: una expedición por la cultura Colombiana" (Crea: an expedition for Colombian culture) was developed.[20] Two cycles of meetings with rural cultural producers and exhibitions took place: one between 1992 and 1994, which also became the basis for a draft of a new law of culture; the second, between 1996 and 1998, culminated with the creation of the Ministry of Culture. The CREA Salon in 1998 presented hundreds of popular artists and dozens of cultists of indigenous, Afro-Colombian, and peasant origin, showcasing their visual, material culture, musical, and dance productions next to "professional artists". The CREA Salon took place next to the 36th National Salon of Contemporary Art (Rojas-Sotelo 2017).[21]

The 1980s and 1990s witnessed the emergence of artists such as (mestiza) María Teresa Hincapíe (1956–2008) with a spiritual, ritualistic, and territorial focus, and Delcy Morelos (1967–) of Embera Katio descent. They opened the space to a contingent of off-center creators (from the provinces), connected to the land, both in their origin and in their artistic practice.[22] The arrival of the Inga artist Carlos Jacanamijoy Tisoy to the art scene, first as an art student at the National University (graduated in 1991) and later as a painter, initiated a chapter for visual artists from indigenous nations in Colombia. His pictorial work has been celebrated and criticized for introducing an indigenous perspective (journeys of yagé) and for its market success for more than two decades.

It is during this time that new forms of art education and the opening of art schools in marginal regions, such as the School of Art at the University of Cauca in 1996, challenged traditional artistic conventions. These events established the possibility of training a whole generation of local artists and intellectuals (breaking with the nineteenth-century tradition). Thanks to its geographical proximity to the Colombian massif and relations with the Regional Indigenous Council of Cauca (CRIC), the Cauca Art School has managed to guide students from indigenous nations (Nasa, Misak, Inga, Cametza, Pastos, Yanaconas, Coconucos, among others) toward contemporary practices, situated cultural activism, and criticism. Other schools are performing similar work in regions of the Caribbean coast and most recently in the Amazon region.[23] These facts (new forms of art education and more art schools) have made higher education accessible to young indigenous people, supporting an emerging generation that represents situated and relational cultural producers in the urban context of arts and letters in Colombia.

On the audiovisual front, following the 1st Continental Film and Video Festival of the Nations of Abya Yala, held in Quito, Ecuador in 1994, indigenous media producers have been producing individually and collectively. Such is the case of the ACIN (Asociación de Cabildos Indígenas del Norte del Cauca), the active communications team of the CRIC (Consejo Regional Indígena del Cauca), supported by legendary filmmaker Marta Rodríguez, which achieved recognition and has given visibility to the lives of contemporary indigenous communities (Villanueva and Guerrero 2013; Rojas-Sotelo 2016). The creation of DAUPARÁ in 2009 (the annual indigenous film festival) has become a showcase of

indigenous cinema and video in Colombia, opening spaces and developing networks for the encounter and dissemination of audiovisual production taking place since the 1970s.[24]

In 1997, the Inga artist Uaira Uaua (Benjamín Jacanamijoy Tisoy, brother of Carlos) obtained a grant to continue his work on the *chumbe* (a weaving belt), a tradition of the Inga (Figure 5).[25] Then, in 2015, he was a finalist for the Luis Caballero Award, the most important recognition for visual artists over 35-years-old in Colombia, for his installation *Auaska Nukanchi Yuyay Kaugsaita: Textile of Own History*.[26] He obtained the Honorable Mention in the VIII Luis Caballero Award. In 2010, the Decolonial Aesthetics project was launched in Bogotá, under the leadership of Pedro Pablo Gómez (Pastos) and Walter Mignolo, with several exhibitions and academic events. The intellectual and artistic project has been sharing ways of thinking about intellectual and cultural production in intercultural and indigenous ways ever since. The creation of peer-review journals such as *Calle 14 Revista de investigación en el campo del arte* (2007) and *Revista de Estudios Artísticos* (2016), both published by La Academia Superior de Artes de Bogotá (ASAB) under the leadership of Gómez, have supported the writing of indigenous cultural production, covering multiple artistic and intellectual practices.

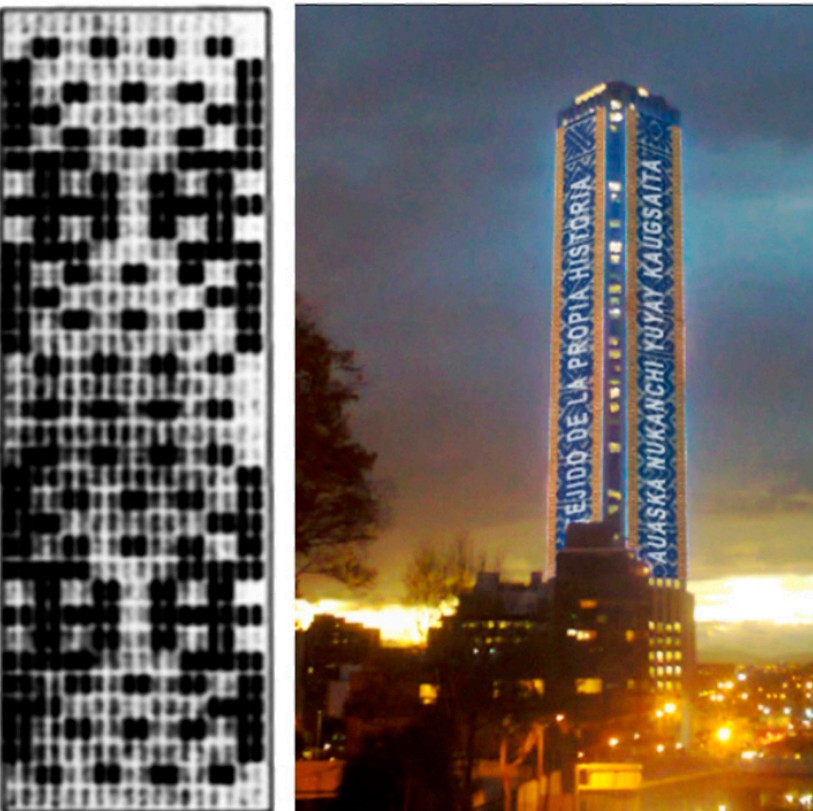

**Figure 5.** Uaira Uaua. Auaska Nukanchi Yuyay Kaugsaita: Textile of Own History. Project for the light system of the Colpatria Tower. Bogotá. 2015. Photo: Benjamín Jacanamijoy.

Since 2014, the University Museum of the National University has developed exhibitions featuring indigenous producers. They have been produced by a transdisciplinary curatorial research model (under the direction of María Belén Saénz de Ibarra) to convincingly present what has been inflicted upon indigenous lands and peoples. Three exhibitions have been developed under the titles "Selva cosmopolita" (Cosmopolitical Jungle) in 2014, "El origen de la noche" (The Origin of the Night) in 2018, and "Conjuro de ríos" (Conjuring Rivers) in 2019.

In 2015, the exhibition "Waterweavers: The River in Contemporary Colombian Visual and Material Culture", curated by José Roca and Alejandro Martín (Roca and Martín 2014), toured some cities in Colombia and the United States, bringing the work of artists, crafts-

men, and designers together (among them indigenous Nonuya illustrator/painter Magaje Guhiu/Abel Rodríguez Muinane) (Roca and Martín 2014).

The National Award for Art Criticism (Ministry of Culture and the University of the Andes) recognized for the first time in 2017 a text that presented the production of indigenous artists, under the title "Soberanía visual en Abya Yala" (Visual Sovereignty in Abya Yala) by Miguel Rojas-Sotelo. In 2018, Miguel Rocha Viva and Miguel Rojas Sotelo co-organized the exhibition "Soberanía Visual" within the framework of the fifth intercultural meeting of Amerindian Literature and Arts (EILA V) in Bogotá (Javeriana University, National Library, International Book Fair). These recognitions and events are part of the collaborative and intercultural work established in 2016 under the label "Mingas de la Imagen", which organizes meetings, workshops, exhibitions, recitals, conferences, and presentations with indigenous and non-indigenous cultural producers twice a year. Lastly, produced by an artist collective, "The Manuel Quintín Lame International Indigenous Salon" began in 2012. It has organized events in 2016, 2018, 2021, and 2022, making visible the work of the emerging generation of indigenous visual producers from the southeast of the country and creating networks for the circulation and debate of their work. The project was founded by Colectivo 83 (led by Edinson Quiñones Falla, a Nasa performer and visual artist). The last section of the article will expand on the exhibitions, Soberanía Visual en Abya Yala, and The Manuel Quintín Lama International Indigenous Salon.

## 3. Embodied Territories/Artists-Territory: Short and Long Temporalities

This section presents a number of issues that impacted indigenous communities in the past century and that still resonate in contemporary artistic practices. Rubber extraction pushed the economic frontier to the deep Amazon, revictimizing communities that had suffered land grabs in the past. Extractive projects are preceded by modern natural expeditions which built ethnographic representation via photography. Both interplayed with violence, which transcended national borders (space) and generations of indigenous people (time). These events bring attention to processes of sovereignty (political and aesthetic) that indigenous people in the Amazon and Andean regions, in particular, have been facing for so long.

In 1909, the London newspaper *Truth* published a testimony (with some photographs), under the title "Putumayo: The Devil's Paradise" by North American engineer W. Hardenburg. It documented how, starting in 1901, the Peruvian merchant Julio César Arana engaged in business with Colombian rubber tappers from *Colonia Indiana* (La Chorrera) to exploit natural rubber for British companies. The exploitation of natural rubber was organized in different sections linked to each other by trails and roads, or by rivers.[27] The publication in the London newspaper unleashed an international scandal and the opening of an investigation into the situation of Casa Arana by the Foreign Office. The British government commissioned Sir Roger Casement, the British consul in Rio de Janeiro (of Irish descent), to investigate the events on the ground. Casement traveled to Putumayo in 1910 and covered a large part of the *La Chorrera* area. He directly interviewed the black workers from Barbados (at the service of the companies) and verified the situation of the indigenous people and the operation of the company. On his return to London, he submitted a detailed report corroborating Hardenburg's claims (Casement 1912). The Indians, according to his testimony, were forced to extract the latex; if they did not, they were punished, flogged, tortured, and murdered. In 1912, the report was published in Great Britain under the title *British Blue Book: Reports of Roger Casement and letters on the atrocities in Putumayo*, accompanied by a series of photographs taken by Casement. In them, the abuse is documented. At the same time, an aesthetic dimension is presented in the bodies and material production of the Uitoto, Bora, Muinane, and Ocaina communities of this region (as a sort of modernist abstract geometry mixed with genocide). Jose Eustacio Rivera's novel *La Vorágine* (*The Vortex*, 1924) also sets the story during the rubber boom. Rubber exploitation returned later in the twentieth century with Richard Evan Shultes, the North American ethnobotanist who for two decades worked to secure rubber for the war

effort while bringing to science thousands of Amazonian species (Davis 1996). These events brought attention to the lives and tribulations of indigenous peoples living on the margins of the new nation-states. Today, at least three contemporary indigenous artists of Uitoto, Bora, and Nonuya ancestry are producing visual representations on the destructive legacies of the rubber boom: Rember Yahuarcani (Perú), Brus Rubio Churay (Perú), and Mogaje Guihu (Colombia). Contemporary films such as *El Abrazo de la Serpiente* (*The Embrace of the Serpent*) by Ciro Guerra in 2015 were also based on such a legacy.

Although Manuel Quintín Lame (1880–1967) did not produce visual materials, it is worth registering the work of this Paéz (Nasa) rebel and intellectual. In his "En defensa de los resguardos" (an official letter dated 17 January 1922) and his posthumous work *Los pensamientos del indio que se educó dentro de las selvas colombianas* (published in 1971), Quintín Lame framed the realities of the indigenous people of his era and called for sovereignty. Quintín Lame was not a man of letters but became Colombia's first and most prolific indigenous activist/writer of the twentieth century. Many of his writings, along with his political performances (sittings, blockages, marches, and symbolic actions), gave continuity to the legalistic practices and documents produced by indigenous subjects during colonial times, who documented and filed claims and petitions before the Spanish Crown. *The Nueva crónica y Buen Gobierno*, written by Guamán Poma de Ayala (1605–1615), *Comentarios reales de los Incas* by Inca Garcilaso de la Vega (published in 1617), and *An Inca Account of the Conquest of Perú* by Titu Cusi Yupanqui (published in 1570, assigned to Fray Marcos García and transcribed by the Inca Martín de Pando) are notable as native versions of the events taking place in the Andean region (Perú in particular). In contemporary times, the work of Antonio Caro (1950–2021) followed a tradition of performance and legalistic denunciation (by using and re-enacting the signature of Quintín Lame), in addition to long-term projects such as *Proyecto Maíz* and *Proyecto 500*. More recently, this approach to cultural production with a clear aesthetic/political stand has been taken by Edinson Quiñones Falla (Nasa), who with an emphasis on issues related to sovereignty works on creating spaces of visibility for local artists of the Andean and Amazonian Piedmont (Colombia and Ecuador). In his personal work, he tackles the persecution of cultural crops such as coca. In his *La Herida Sana y la Cicatriz Queda* (Figure 6), Quiñones Falla tattooed the image of the god of coca on his back and then removed it from his body to show the wound and scar of such practice on the land of his ancestors. He presented the piece of skin (in a frame) as the document of the action. In his collective research/action, Quiñones Falla produces exhibitions, documentaries, and events.

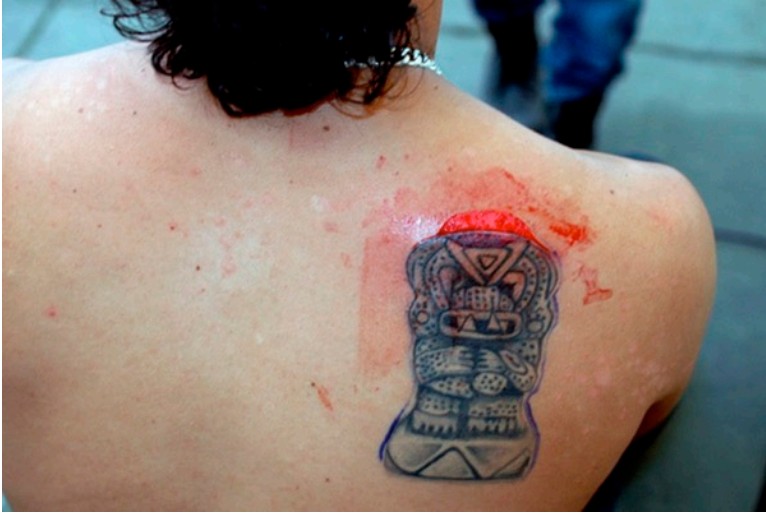

**Figure 6.** Edinson Quiñones Falla. *La Herida Sana y la Cicatriz Queda: Escarificación Dios de la Coca* (*The Wound Heals, The Scar Remains: Scarification God of Coca*). Performance. 2006. © The artist.

The construction of the image of the indigenous has been in the hands of the white gaze. In Rivera's *La Voragine* (1924), indigenous images (descriptions) are included amid the horrors of rubber extraction. In Jorge Issacs' novel *María* (1867), the author introduces the short love story of Nay and Sinar, apparently indigenous of Nasa and Misak origin, as a framing for the tragic love story of María and Efraín (the Nasa and Misak had historical conflicts as in the archetypical Romeo and Juliet story). However, the most important works in the form of illustrated ethnography were published in Berlin in 1910–11 and 1917 by Theodor Koch-Grünberg, titled *Zwei Jahre Unter Den Indianern. Reisen in Nord West Brasilien, 1903–1905* (*Two Years Among the Indians. Travels in North-West Brazil, 1903/1905* 15 June 2023 *From Roraima to the Orinoco*). He produced the most complete archive of the peoples of the Brazilian, Venezuelan, and Colombian Amazon at the time. His entire archive and collections are kept at Berlin's Ethnographic Museum. Koch-Grünberg returned for a third expedition (1921–1923), with pioneer Brazilian filmmaker Silvino Santos Silva. As a result, he died of malaria in 1924. In 1926, Konrad Theodor Preuss published a *Visit to the Kágaba of the Sierra Nevada de Santa Marta, Parts I and II*. The complete investigations of Preuss (who lived in Colombia during War World I) cover the Kogui of the Caribbean to the Uitoto of the Amazon. A posthumous work, *Religion und mythologie der Uitoto* (*Religión y mitología de los uitoto*), was published in Germany in 1938. That volume includes numerous mythical–ritual texts, not published in Colombia until 1993 and 1994. Koch-Grünberg and Preuss were typical ethnographers who described places and people, gathered narratives and songs, and illustrated, photographed, and collected cultural material to further their anthropological and ethnographic studies (taking many treasures at times without permission). Today, the work of artists such as Rosa Tisoy Tandioy (Inga, Putumayo) has been turning the camera inwards as counter-ethnographic statements and counter-portraits. In her "Sara Indi" series of self-portraits, she draws with corn on her body, revealing what is behind the sacred writings of the chumbe. In her "Kanimi Alli Uiñangapa" (digital interventions), she embodies territory, becoming one she-earth (Figure 7). In Tisoy Tandioy's portrait series, there are no faces but bodies-territory where seeds (corn) and wombs situate a cosmovision that is rooted in local practices (Figure 8). The performance pieces and documentation by Tirsa Chindoy (Inga), "Hilando Memorias" (2016), and Juliet Morales (Mizak), "Resistencia" (2019), are also self-ethnographies (Figures 9 and 10). There, the role of indigenous women is tested, and the production of media pieces (video installations, photos, and video) establishes new archival practices based on women's activities within their communities. Finally, the audiovisual work of Keratuma Domico, "Mudrua" (2012), and Olowaily Green, "Galo Dugbis" (2020), also turned the camera in(side), producing biographical/collective explorations of the return and re-telling of original stories from the perspectives of women (Figure 11).

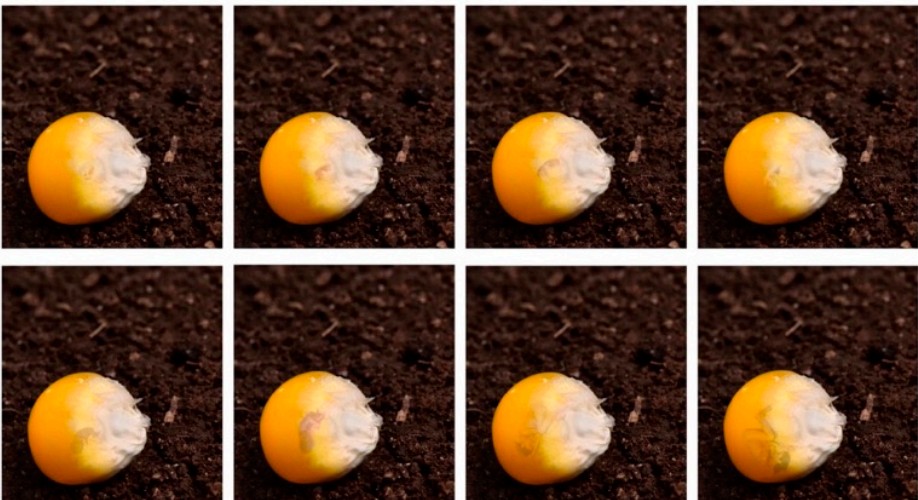

**Figure 7.** Rosa Tisoy-Tandioy. *Kanimi Alli Uiñangapa (soy buena semilla)*. Digital photography. 2013. Photo: Rosa Tisoy-Tandioy.

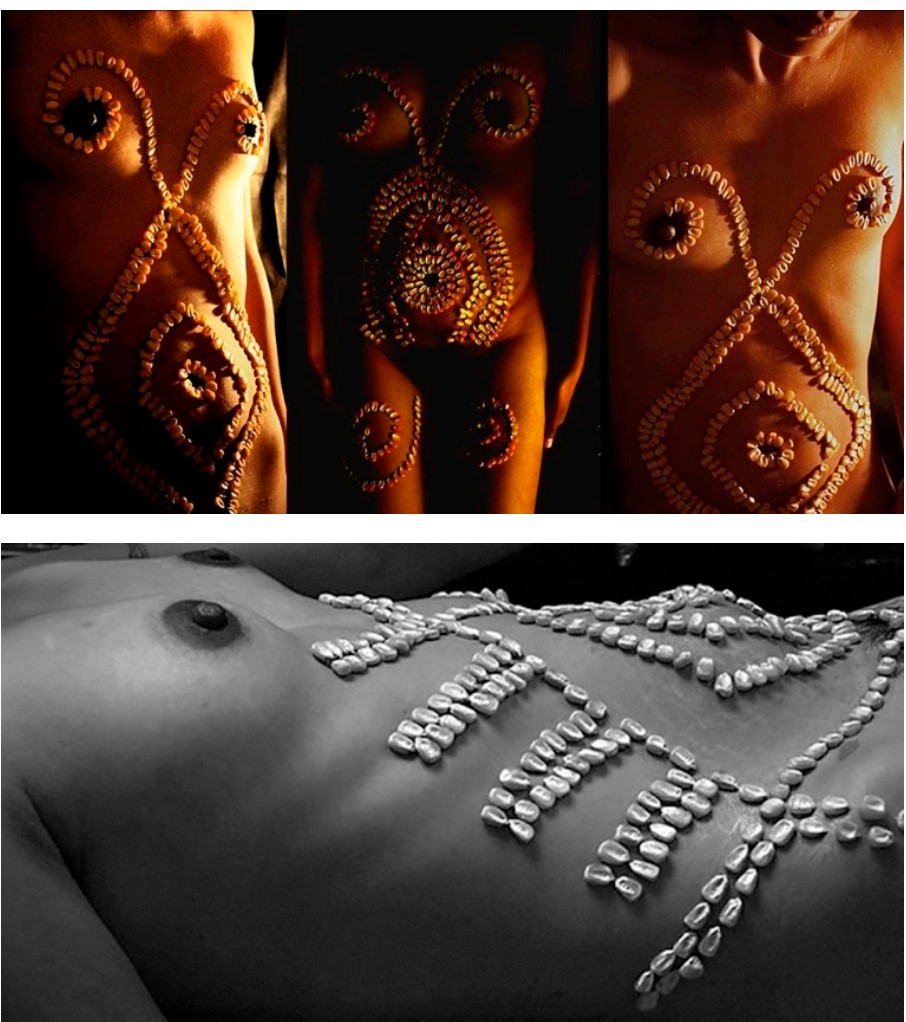

**Figure 8.** Rosa Tisoy-Tandioy. *Sara Indi (conexión cuerpo-sagrado maíz)* Series. Photography, 2009–2014. Photo: Rosa Tisoy-Tandioy.

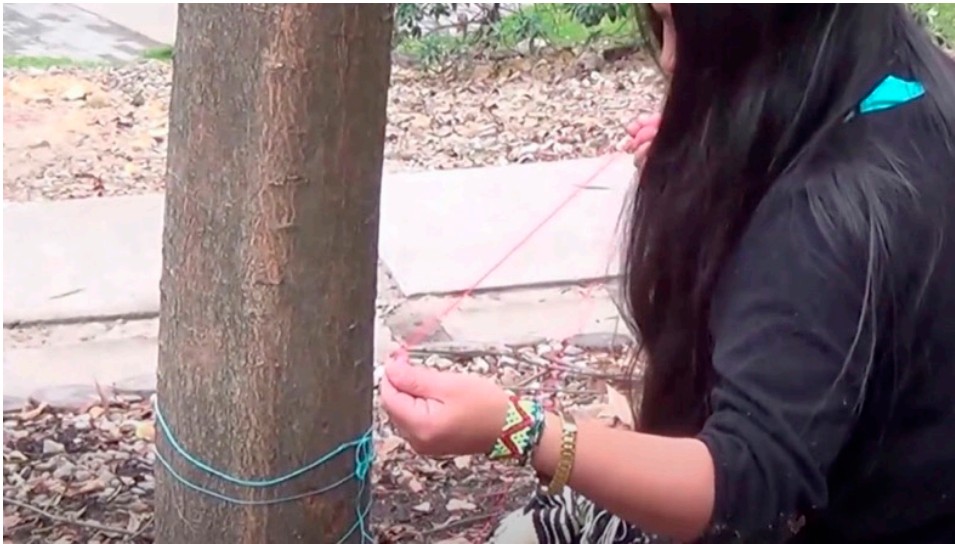

**Figure 9.** Tirsa Chindoy. *Hilando Memorias* (performance and video), 2016. Photo: the author.

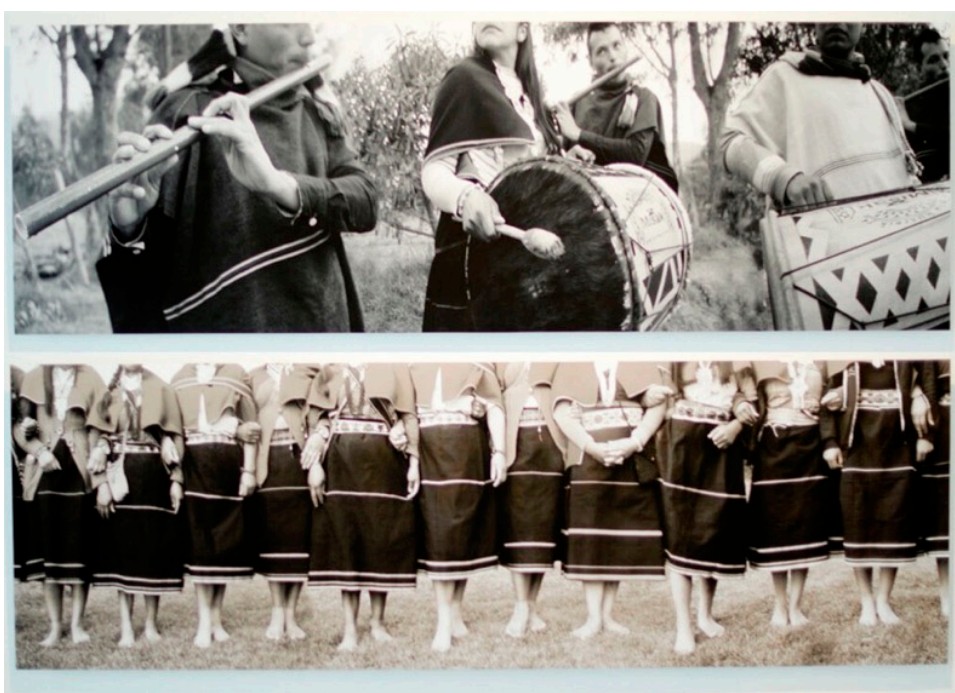

**Figure 10.** Juliet Morales (Mizak). *Resistencia* (performance and photo installation), 2015. Photo: Juliet Morales.

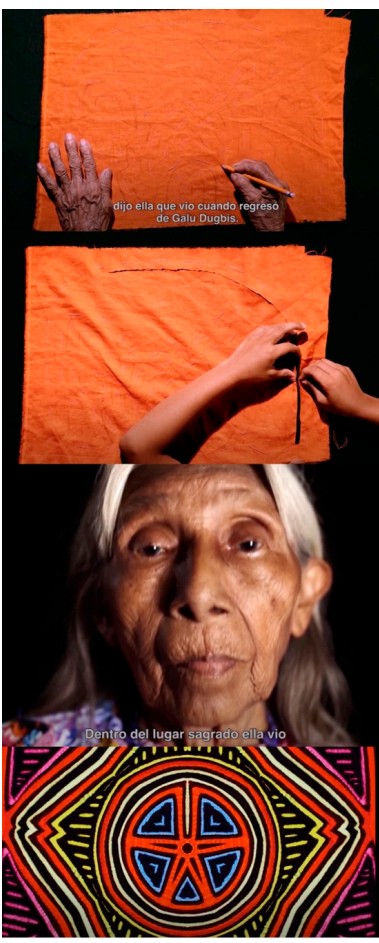

**Figure 11.** Olowaili Green (Guna Dule). *Galo Dugbis* (video), 2020. Stills from video.

## 4. Fabulation Becomes Realization: Spatialization, Visual Sovereignty, and the Manuel Quintín Lame Indigenous Salon

Artistic practices such as the ones mentioned above are taking place in indigenous territories (many in dispute), the urban public space, as well as in galleries and museums. Because of their dynamics, embodied-territory producers move in between worlds, envisioning new stories of their constitution. The demounting (tackling-down) of historical monuments of European colonizers, by young people of indigenous communities (the Mizak in particular) in cities such as Popayán, Cali, and Bogotá, during the social uprising of 2019–2021, not only constituted a symbol of social, political, and historical vindication, but are also epistemic awakenings at the center of the process of self-education and a critical revaluation of the concept of history. These actions, which are not new (let us remember the toppling of Diego de Mazariegos in San Cristobal, Chiapas on 12 October 1992), respond to the erasure of indigenous stories in the construction of modern nations. On 16 September 2020, a group of young Mizak and Nasa demounted the equestrian statue of Sebastián de Belalcázar on "El Cerro Tulcán" in the outskirts of the city of Popayán. The sacred mount (an earth pyramid) was a ceremonial burial for centuries. It was used to create an equestrian monument to the "founder" of the city (of Popayán and Cali) by white authorities in 1940. In 2015, Juliet Morales and Manuel Muelas performed a piece titled "Lo vamos a sembrar" (we are going to plant it) on the grounds of the monument (Figure 12). A call to action preceded, in fabulist terms, the historical judgment that supported the demounting of the statue in 2020 (Rojas-Sotelo 2023). On the one hand, judicialization and payment for damages to public property are requested (the indigenous participants in these actions are branded as savages and uncivilized, once again); on the other, the events serve as a trigger for debate and an initial form of restitution for historical debts, for the whitening of history, and for the repression and denial of basic rights. The actions, although controversial and of an iconoclastic type, were peaceful and created spaces of debate about history, representation, and identity. They can be considered forms of radical horizontality, challenging the verticality of political and historical discourses (2023: 165). Many other monuments have fallen, and many others will fall in search of sovereignty (Figure 13).

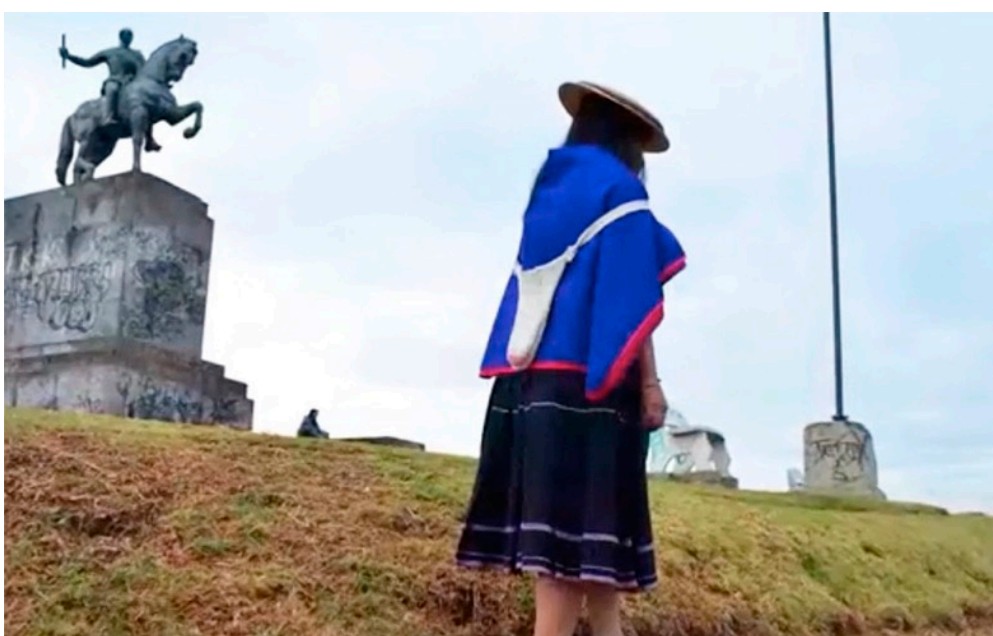

**Figure 12.** Juliet Morales and Manuel Muelas. *Lo vamos a sembrar* (performance of planting coca seeds around the monument), 2015. Photo: Juliet Morales.

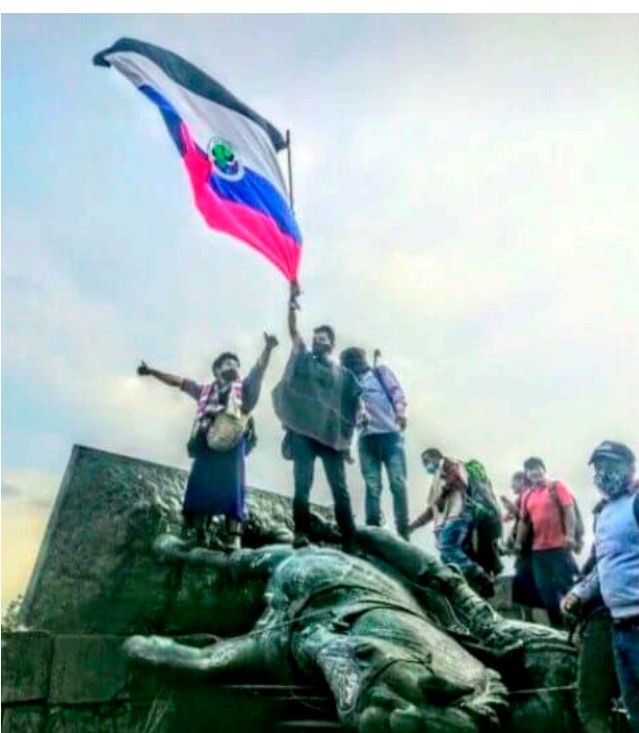

**Figure 13.** The Equestrian Statue of Sebastián de Belalcázar falls. Sep. 16, 2020. Photo: Martha Peralta.

Visual Sovereignty was an exhibition that took place in April–June 2018, as part of EILA V, organized by Miguel Rocha Vivas and Miguel Rojas-Sotelo, and co-funded by Universidad Javeriana in Bogotá. The conceptual script of the show was based on an essay of the same title that won the 2016–2017 national prize on art criticism in Colombia. In the essay, as well as in the exhibition, an ecocritical and epistemic reading of a group of cultural producers and their practices was explored to expand the horizon of avant-garde aesthetic canons. It featured seventeen participants to establish multiethnic, multigenerational, and intercultural bridges.[28] The exhibition functioned as an extended territory where cardinal points structured the exhibition space (east–west, south–north). For example, pieces such as "Chingaza", an installation of seven thousand golden $100.00 peso coins by Jeisson Castillo, who wrote the word *chingaza* (a sacred highland moor/páramo) on a wall toward the east. Chingaza, which means in the muisca language "mountain of the night good", faced a piece titled "Maiz" (five tons of yellow corn) by Carlos Uribe, establishing a visual, aesthetic, and material dialogue (páramos give water to the plantations located in the high plateaus of the Andean range) (Figure 14).

Other pieces, such as the ritual work "Anaconda para Chingaza" by Dioscóredes Pérez (an elder performance practitioner), established the riverbed and symbolic flows for the paintings (on llanchama, an Amazonian tree bark, and on canvas) by Brus Rubio Churay and Taita Domingo Cuatindioy (an Inga shaman and visual producer) (Figure 15). Their work always concerns the flows of history, memory, and the water of rivers that are born in the Andes and end in the Atlantic Ocean (west–east).

On the south–north axis, textiles, ceramics, wood carvings, illustrations, and digital prints were on display. On the south, pieces of the Tropenbos collection of Amazonian material culture extended into the gallery. It is worth noting that the idea to anchor the arts on traditional practices and figurative languages situated the works on exhibition beyond the usual parameters set for contemporary art. In this section, the pieces are identified by their place of production: ceramics from Puerto Cordova, Amazonas; anaconda wood-carving benches and *balayes* (weaving pieces for daily uses) from the Vaupes and Vichada; fishing traps and illustrations of the cycle of floods by Mogaje Guihu from Peña Roja in Caqueta; Maguaré drums from the Uitoto communities of the Amazon, among others

(Figure 16). On the north axis, the work of industrial designer and visual producer Juan Obando recreates another forest. "(New) Cartel" (2010) builds a virtual landscape made from a repetition of a pattern of oil palms and large weapons (Figure 17). The piece visually tells the expansion of the economic frontier via violent tactics by organized crime to hold indigenous territories by force. The redrawing of the territory consolidates criminal routes and capitalizes on the green economies of mono-crops such as oil palm in the Atrato region of Choco and Antioquia territories of the Wounan and Embera in the north–west, near the border with Gunadule of Pánama. Visual Sovereignty presented an expanded horizon of cultural production in dialogue with contemporary issues and artistic practices in an exhibition space.

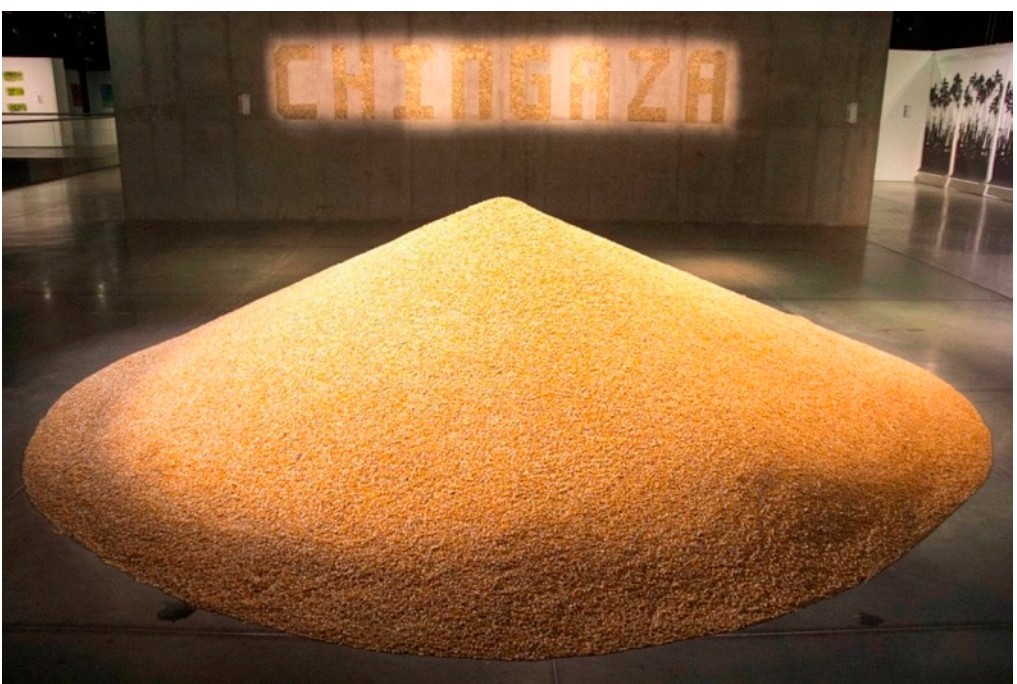

**Figure 14.** Jeisson Castillo, *Chingaza*, 2018 (golden coins on the wall) and Carlos Uribe, *Maiz*, 1994–2018 (five tons of yellow corn on the floor). East–west axis, Exhibition Hall. Soberanía Visual, 2018. Photo: Jeisson Castillo.

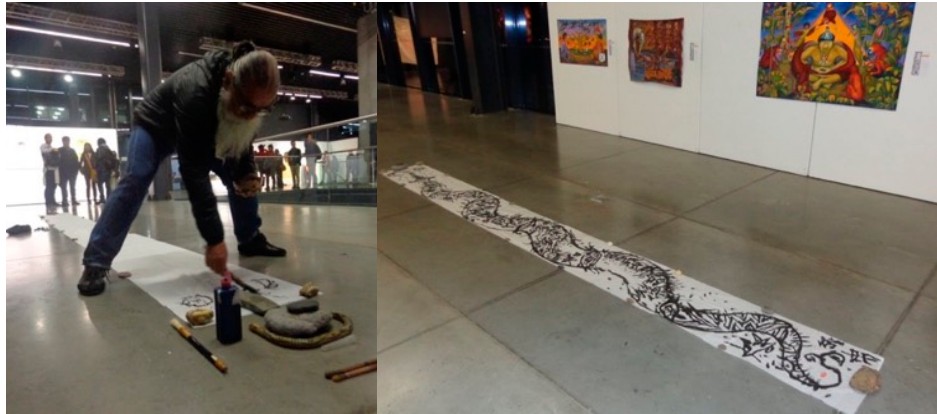

**Figure 15.** Disocóredes Pérez, *Anaconda para Chingaza*, 2018 (performance drawing on rice paper, ink, and rocks). In the background, Brus Rubio, *La Concentración*, 2017 (right) and pieces on llanchama bark. Soberanía Visual, 2018. Photo: the author.

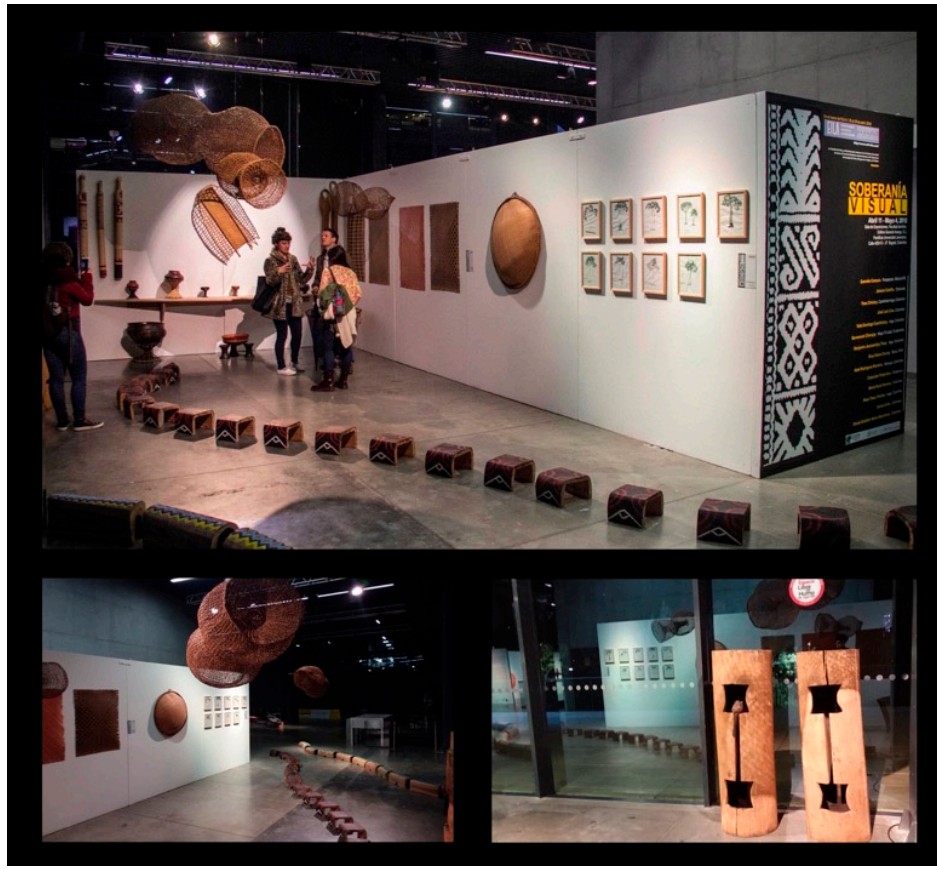

**Figure 16.** Tropenbos Collection of Amazonian material culture. Soberanía Visual. Exhibition Hall, Universidad Javeriana. 2018. Photo: the author.

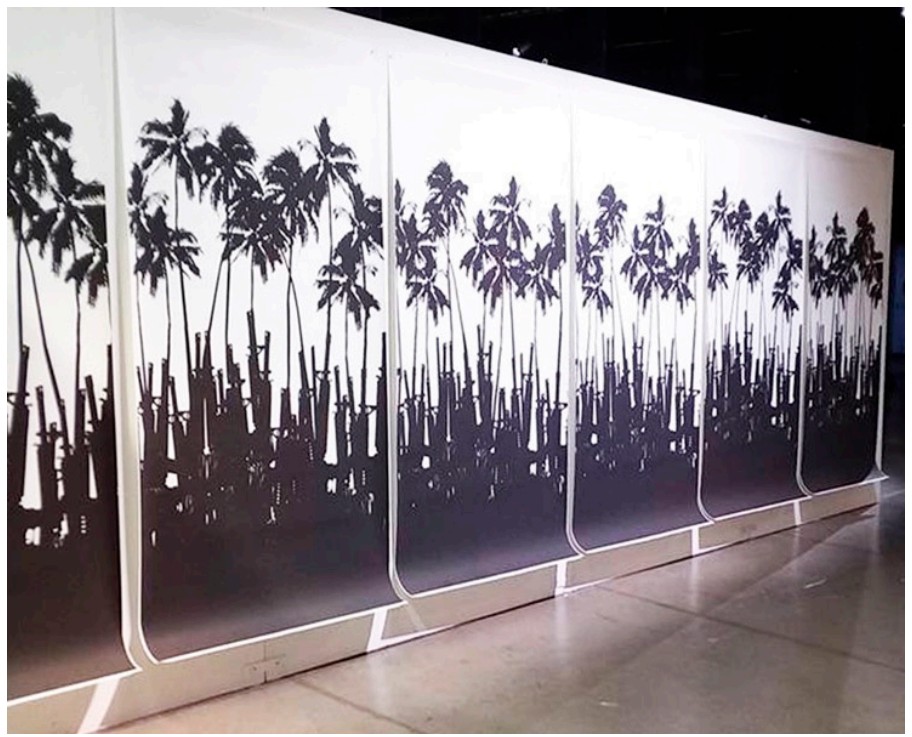

**Figure 17.** Juan Obando, *(Nuevo) Cartel*, 2010. [Digital print on paper. 120 cm × 240 cm each]. Soberanía Visual. Exhibition Hall, Universidad Javeriana. 2018. Photo: the author.

Finally, what started as an exhibition project, the *Salón Internacional de Arte Indígena Manuel Quintín Lame* (International Indigenous Salon Manuel Quintín Lame) in the city of Popayán (Cauca) has evolved into a scholarly/curatorial practice. Initially organized by the Colectivo 83 led by Edinson Quiñones Falla (Nasa), it now functions under the label *Minga de pensamiento y prácticas decoloniales* (Minga of Decolonial Thought and Artistic Practices).[29] Working from the heart of the indigenous resistance in Colombia, the Cauca region, the collective had capitalized on their location and meaning. On the one hand, it has given distance and autonomy to the group; on the other, the event has grown under the tutelage of a situated art school within the University of Cauca (with a long tradition of studying the indigenous struggles of the region as well as educating new generations of contemporary cultural producers). The Indigenous Salon, at first, started as a conventional call for participation, exhibition, and academic events; later, it became a curatorial space for the redefinition of indigenous life in the territory, including a residence program (Papayork, in the city of Popayán) (Figures 18 and 19). Its goal from the start was to become a decentered and contextual voice for the production of young indigenous cultural producers. The collective won the Visual Arts Laboratories Stimulus Scholarship (Ministry of Culture 2016), which generated an academic space to demonstrate the differences and similarities of the cultural and artistic manifestations of the local indigenous populations. In addition, Quiñones Falla has curated and co-organized representations of artists from Cauca for the Arco fair (Madrid 2015), the *16 Regional Art Salon*, Pacific zone (2016), as well as the exhibition *Imagen Regional 7*, southwest zone (2021–21), for the Ministry of Culture and Banco de la República. The collective organized its first show in 2012, with editions in 2016, 2018, 2021, and 2022. Such visibility opened opportunities for the collective to travel across the country and abroad, exchanging with other individuals and collectives. This has allowed them to bring the salon to cities such as Cali and Bogotá (Museo de Arte Contemporáneo and Universidad de los Andes art gallery), expanding their reach and their network (Figures 20 and 21). In 2020, the collective regrouped under a new name, the *Minga de pensamiento y prácticas decoloniales* (Collective of Decolonial Thought and Artistic Practices) (note 29).

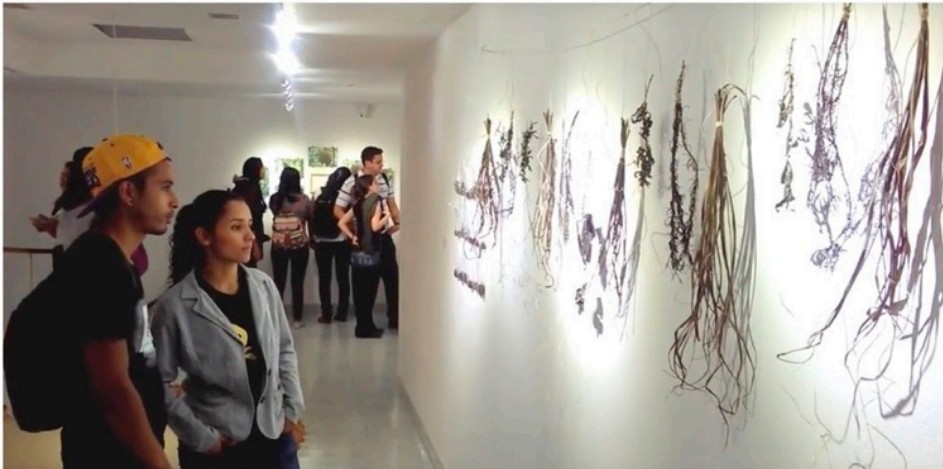

**Figure 18.** Primer Salón Internacional de Arte Indígena Manuel Quintín-Lame, Museo Negret, Popayán, 2016. In view: Rosa Tisoy-Tandioy, *Tinii*, 2016. Photo: Rosa Tisoy-Tandioy.

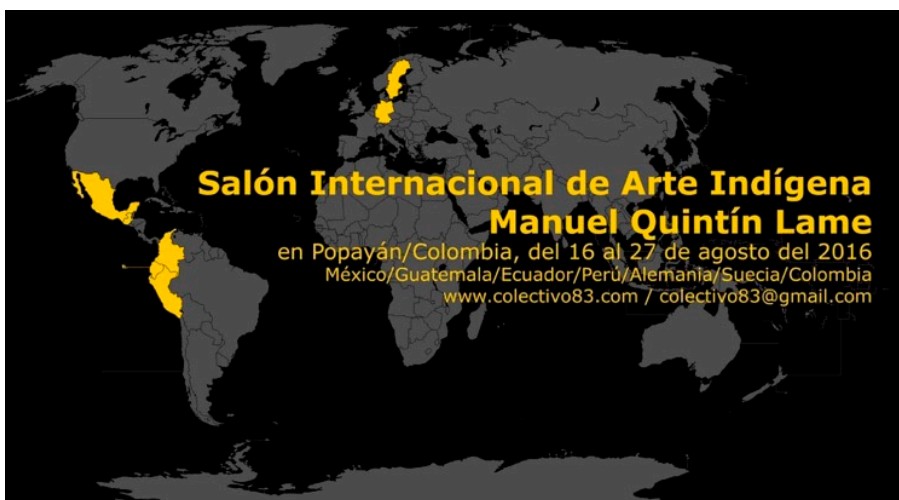

**Figure 19.** Primer Salón Internacional de Arte Indígena Manuel Quintín-Lame, Museo Negret, Popayán, 2016. Digital poster.

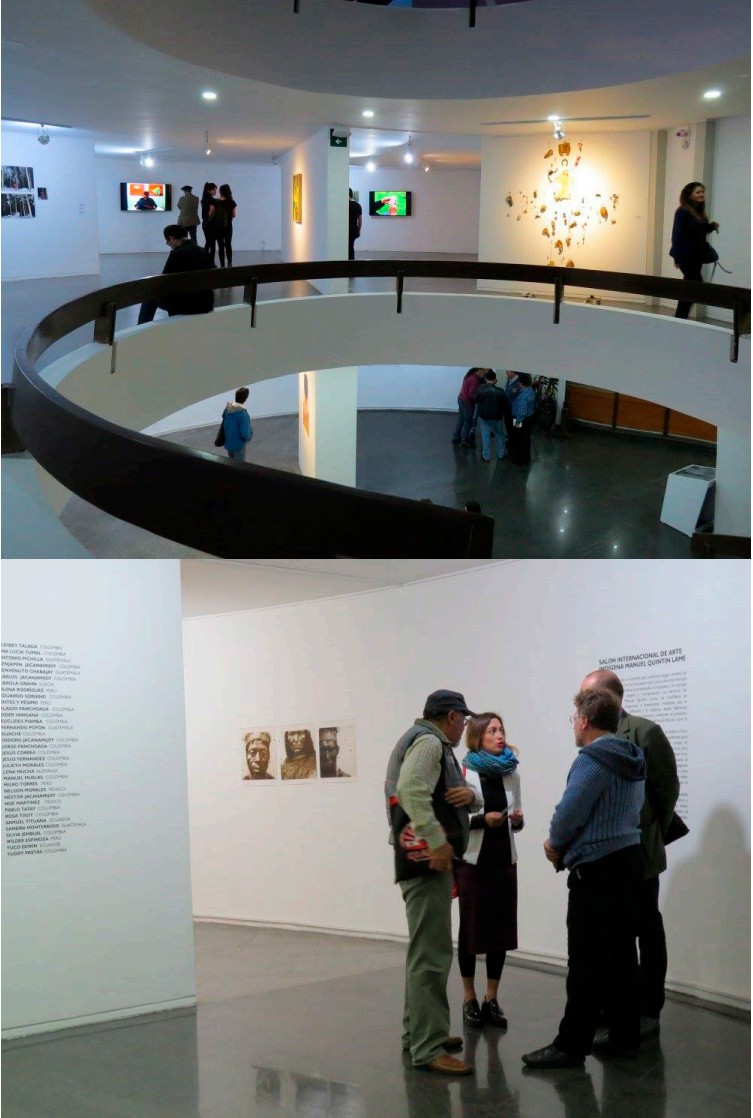

**Figure 20.** Salón Internacional de Arte Indígena Manuel Quintín Lame, Museo de Arte Contemporáneo, Bogotá, 2017. Photo: Edinson Quiñones Falla.

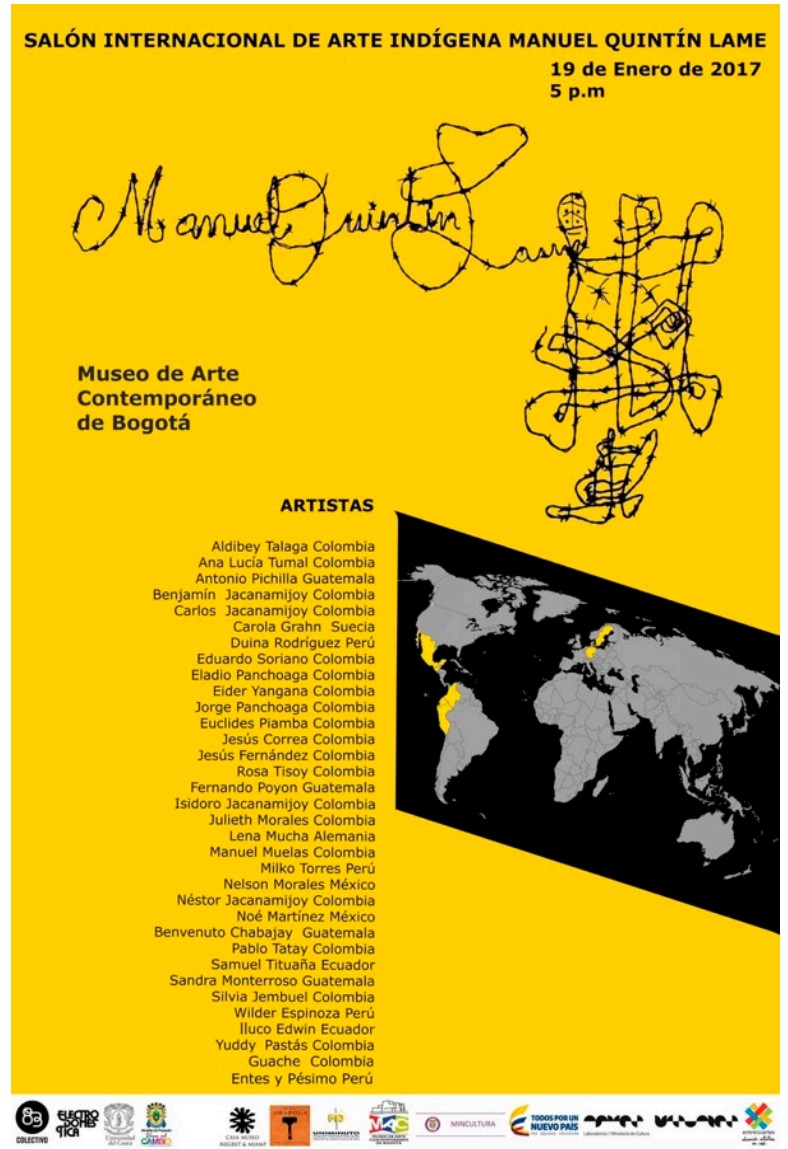

**Figure 21.** Poster, Salón Internacional de Arte Indígena Manuel Quintín Lame, Museo de Arte Contemporáneo, Bogotá, 2017. Digital poster.

Under the new structure, the collective has been developing documentary/experimental videos and community workshops to tell stories in a decolonial key. By emphasizing issues such as environmental degradation in their territories (due to illegal crops and uses of energy and water resources), the collective has taken a more scholarly, research-based, and community-oriented approach. In their video *Mama Yuma's Womb*, presented at the 46th National Salon in 2022, they developed a manifesto message from freshwater to saltwater. For the 22-min video, the collective traveled through the territory, from the birth of the so-called Río de la Magdalena/Yuma (at paramo de la papas), documenting the relationship with indigenous water guardians of the Yanaconas, Coconucos, Mizak, and Nasa communities, to the delta of the river on the Caribbean coast. In their piece *Resignification and Collective Construction of Decolonial Imaginaries* (video, 60 min, 2020–2021), the collective reflects critically on the recent history, using academic research and the living memory of the Nasa and Inga peoples. The piece (produced by Estefania García, Phuyu Uma, Eyder Calambás, and Miller Muñoz) was presented at the Unam Biennial 2021–2022. It looks at the legacy of Guillermo León Valencia, a modernizer president of Colombia (1962–1966) originally from the white elites of the city of Popayán (Cauca), who crafted a "total war" policy against indigenous communities. León Valencia attacked, using the

army, indigenous organizations under the argument that they were creating independent republics, extending the genocide, repression, and hostilities of the colonial period to the contemporary moment. By deconstructing such stories and by bringing the voices, images, locations, and temporalities of indigenous survivors, the collective is moving beyond an aesthetic statement into the realm of rewriting history (Figure 22).[30]

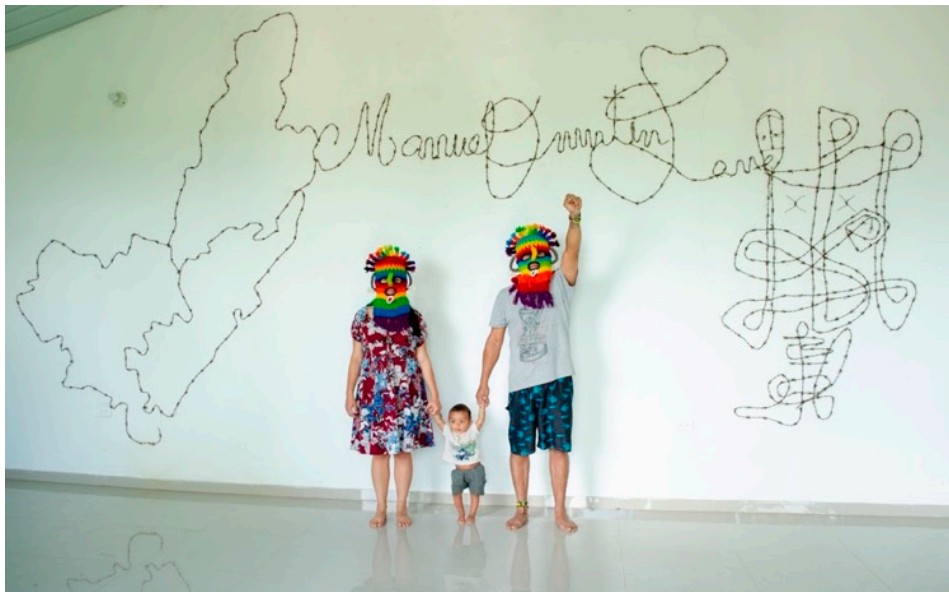

**Figure 22.** Estefanía García Pineda and Edinson Quiñones Falla (and son). Minga de Prácticas Decoloniales. Popayán, 2023 (drawing on barbwire. Map of the Gran Cauca and Manuel Quintín Lame's signature). Photo: Eyder Calambás.

The collective, led by Quiñones Falla, continues to work for a solid regional presence of artists-territory that enter and leave the art system and generate local creative processes. They summarize their current work as follows:

> "Una práctica de creación contemporánea desde el pensamiento originario, debe ir equilibrada con la planeta, porque es el útero, es la madre, cruzar la frontera epistémica para llevar nuestros creaciones y sentires a un cubo blanco es un acto de orden pedagógico ecológico político". Minga de Practicas Decoloniales. 1 February 2023[31]

## 5. Conclusions

To rethink contemporary Latin America Art, it is necessary to look critically at the historical progression and the cultural dependency on Eurocentric and North American models. By using alternative ontologies (other ways to be in the world, mentioned here as fabulations and myths) such as the one expressed in the Tree of Abundance, it is possible to see cultural production from coloniality, rethinking art historical models, and allowing other constellations to be part of discourses of knowledge production and world-making. The Tree of Abundance spreads its branches and roots, covering, giving shelter, shade, food, good air, and supporting the land we inhabit. The tree is a metaphor for the forest that is not seen due to vertical and partial constructions of history, culture, and art detached from the everyday and from the historical experience of the indigenous peoples of Abya Yala. The Tree of Abundance recognizes conditions of representation in the territories for a time immemorial. The Tree of Abundance is not a methodology but a metaphor (allegory if we must) of the richness of cultural expressions within the multiple indigenous nations in Abya Yala. The myth behind the tree has survived as an origin story, passed by oral means and incarnated via ritual across generations, The Tree of Abundance announces an alternative ontology parallel to the Western one, now being enacted thanks to the situated work of

people such as Mogaje Guihu and Brus Rubio Churay, and a generation of indigenous and intercultural subjects and collectives. The metaphor explains how even after contact (the falling of the tree), a traumatic event that terraformed the continent (and by extension the planet, via coloniality), a subtract of alternative ways of living and doing reminds in indigenous communities. Debates and political and intellectual work during the 1970s and 1980s before the 1992 anniversary of the contact/clash renamed the continent and brought conditions for the expansion of rights of nation-states (for indigenous peoples) for cultural production. In the 1990s and 2000s, direct action and resistance by communities (Zapatistas, the Indigenous Guard of Cauca, among others) pushed for declarations and bills of rights by multilateral organizations (ONU, OAS). These, among other events, created conditions for the emergence of a new generation of embodied territories. Cultural producers build on long genealogies of sovereign representation, responding with a wide range of contemporary means (visual, textual, bodily, and multimedia) to issues that still affect their communities (land grabs, resource extraction, racialization, marginality, etc.).

Adaptation, resistance, and re-existence occur when embodied territories recognize contextual realities (time), location (space), and forms of repression and liberation (action) within coloniality. These cultural makers, embodied territories, work in particular historical contexts and situations, react in multimedia fashion to long stories of repression, and produce individually and collectively. With their actions, they seek to open other interpretive categories, such as in the renaming of the territory (Abya Yala), an epistemological as well as a political act. Some work in poetic ways, writing on the body with seeds, such as in the case of Rosa Tisoy Tandioy; by re-centering female indigenous narratives, as in the performances by Tirsa Chindoy and Juliet Morales; or via audiovisual means, such as Keratuma Domicó and Olowaily Green. Also, new dialogue is possible as seen in the actions of radical horizontality achieved by demounting monuments of conquerors, those that took the tree down in the first place (Figure 23). They all attest to the abundance by producing events, actions, callings, establishing relations and collaborations, such as in the case of the Minga de Prácticas Decoloniales, among others.

By recognizing long temporalities, historical presence, ethnic violence, racialization, nature, and gender abuse, these embodied territories bring and re-inscribe experiences that are not part of the modern archive in art history. In the process of rethinking Contemporary Latin American Art, a sort of process of return happens. On one hand, there is a realization that this is not only a geographic space constructed by taking down a constitutive ontology and replacing it with another (the modern/colonial) but also a sociopolitical, cultural, and epistemic territory forged by coloniality. On the other, the contemporary era did not begin around 1950, after WWII, when humanity began to witness unprecedented global exploitation of both vertical and horizontal frontiers (the great acceleration), but much earlier. It is necessary to break the linearity of historical Western time and sync the cultural experience and production in cycles of return.

A debate about how contemporary art has a responsibility in the construction of restorative historical memory and its presence in public spaces has been opened. With these acts and insertions into the public space, the gallery, the museum, publications (such as this), etc., a horizon of intercultural dialogue can write a more inclusive and democratic history, one that builds not on the ruins of indigenous cultures but on their rebirth. Individual and collective creative work embodies and incorporates the possibility that rises and joins social discontent. This abundance, now expressed by a new generation of indigenous makers (also allies) across the continent, sees how the prevailing social system does not correspond or give space for social transformation, justice, and full participation in the nation-state and acts upon the injustices of the system with determination. These poetic, political, and aesthetic forms and actions are not new; they invoke all those undermined and silenced by coloniality. A sort of symbolic justice is achieved by reclaiming visual sovereignty.

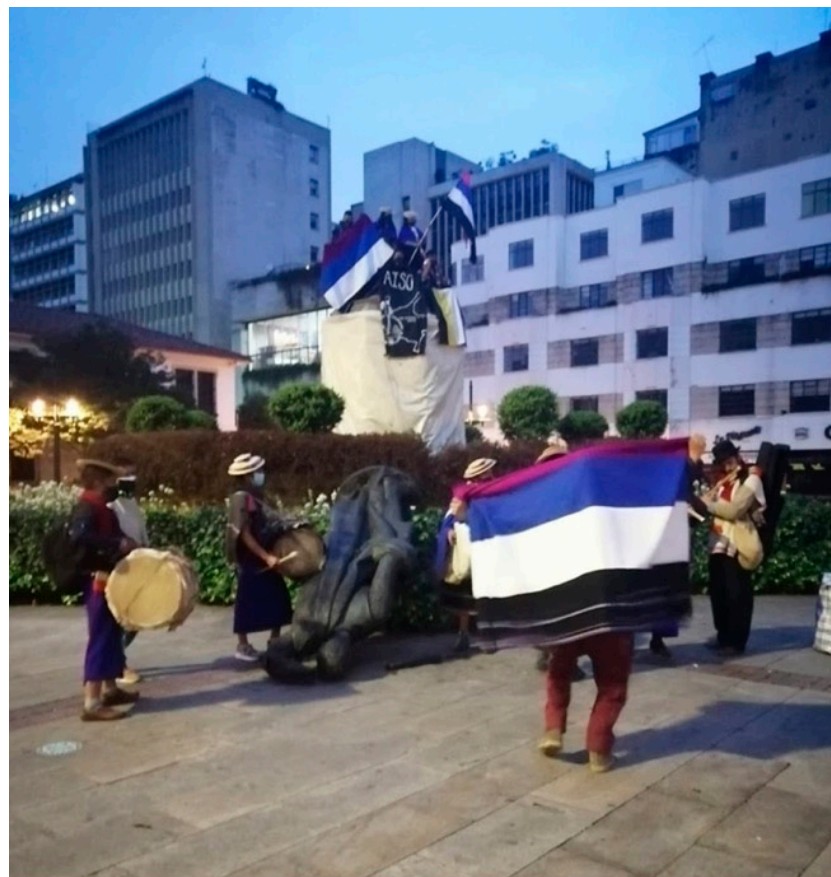

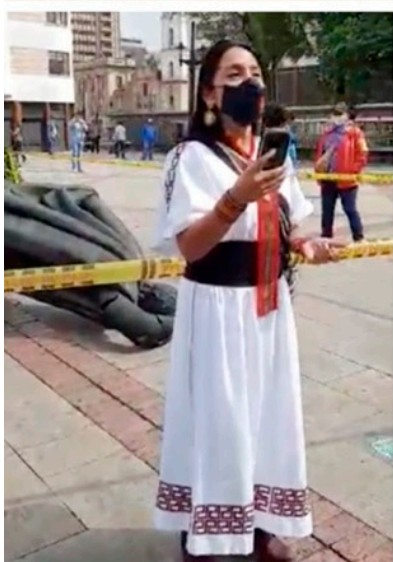
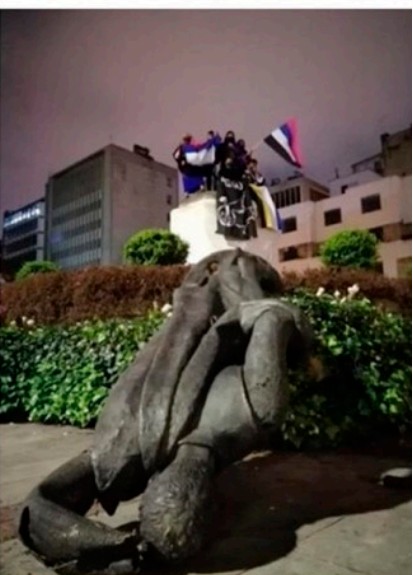

**Figure 23.** Indigenous Mizak and Arhuaco youth after demounting Gonzalo Jiménez de Quezada's statue in downtown Bogotá on 7 May 2021. Photo: Daniel Martínez.

**Funding:** This research received no external funding.

**Data Availability Statement:** Not appliable.

**Conflicts of Interest:** The author declares no conflict of interest.

## Notes

1.  Since the canonical texts *The Death of the Author* (Barthes, 1967) and *What is the Author?* (Foucault, 1969), the function of the concept has been deployed in analyses and critically dismantled within a wide range of art practices and the art system. Authorship has also been deconstructed by feminist and decolonial theories in the past decades, although it persists mostly due to the forces of the market and the limits of decentralized cultural theory in Art History. For a contemporary discussion about the issue see Hammam Aldouri (2021).

2.  On Decolonial Aesthetics see Mignolo, W. 2010. "Aiesthesis decolonial", *Calle14* 4(4): 10–25; Mignolo, W. 2010. Que es estética(as) decolonial(es)? https://www.youtube.com/watch?v=znaaLQZOb0g&t=2s (accessed on 8 December 2022); Albán Achinte, A. 2011. "Estéticas decoloniales y de re-existencia: entre memorias y cosmovisiones", *La arquitectura del sentido II*. 87–117; "Decolonial Aesthetics (I). The Argument As Manifesto / Estetica decolonială (I). Un argument ca manifest". 2011. *IDEA* 39: 89–97 pp. 89–97; Mignolo, W. and Pedro Pablo Gómez (eds.). 2012. *Estéticas y opción decolonial*; Mignolo, W. and Rolando Vázquez. 2013. Decolonial AestheSis, *Social Text Online*, eds. 3. Dossier. Rojas-Sotelo, M. 2014. Decolonizing Aesthetics/Aesthesis. *Oxford Encyclopedia of Aesthetics*. Oxford University Press. Etc.

3.  This emergence is taking place across the continent, Mexico, Guatemala, Brazil, Argentina, Perú, and Chile in particular. This text looks with more detail into what is happening in the indigenous territories of what is considered Colombia, due to its location as a connector of South and Central America. In the analysis, political borders are at times fluid since spaces such as the Andes or the Amazon have commonalities and go beyond national borders. The same happens when dealing with issues that traverse such political divisions (such as resource extraction, violence, memory, etc.). Indigenous art takes many forms and ways of being presented, circulated, and received. See, for example, the newly created (2019) Museo de Arte Indígena Contemporáneo (MAIC), as a touring exhibition complex in Mexico with headquarters in Morelos, https://www.mexicoescultura.com/recinto/67292/museo-de-arte-indigena-contemporaneo.html (accessed on 8 December 2022); or the emergence of indigenous art in Brazil, as presented by cultural critic Niagara Galarraga C. 2023. "El arte indígena conquista los museos de Brasil". *El País*. 8 January.

4.  In his book *Scarcity and Frontiers: How Economies Have Developed Through Natural Resource Exploitation*, Edward Barbier (2012) defines the Contemporary Era. The author argues that it began around 1950 when humanity witnessed unprecedented global exploitation of both vertical and horizontal frontiers, with much of this expansion occurring in the developing regions of the world after decolonization.

5.  Latin American Contemporary Art is a subaltern, emerging under the shadow of the Cold War and the political, cultural, and economic dependence of the United States and Europe. It also carries the scars of its more than three hundred years of European colonial history and coloniality (in the form of neoliberalism in the contemporary era) as a constitutive force.

6.  On the MOMA exhibit, critic Thomas McEvilley interpreted the exhibition as projecting modern Western ideas of art and the individual artist onto cultures for which art did not exist as a separate category and in which formal change could not be credited principally to individual artists. *Magiciens de la terre* intended to create a plain field in which 50% of participants were non-Western, creating a fake sensation of fairness. The 1991 Habana Biennale faced fierce debates about the legacies of coloniality, the modern and the traditional, and showed side-by-side multiple works from competing visions of the art world in a general essay that brought artists from three continents together for the first time gathering in Havana.

7.  It was documented that in the Consejo Mundial de Pueblos Indígenas (World Indigenous Council) of 1992, the term *Abya Yala* was admitted in replacement of América. In gunadule (kuna) language, Yala means "earth/soil" or "territory". Abia means "hole of blood", "mature mother", "mature virgin", or "land in full maturity". It was Aymara leader Takir Mamani from Bolivia who proposed that all indigenous use it in their declarations and documents. Mamani had traveled in the early 1970s to Pánama, where he got to know the term; in many meetings of indigenous intellectuals during the 1980s, this was shared. In 1992, he said "to name with a foreign term our cities, towns, and continent, equals to keep our identity under the will of the conqueror and their inheritors". Costa Rica, 1992.

8.  In other regions of the continent, renaming is also taking place, e.g., Pindorama instead of Brazil or the Turtle Island to name the indigenous nations of the north (from the Great Lakes to the Arctic).

9.  The indigenous EZLN movement was an eye-opening event for both the Mexican government and the non-indigenous population to realize the alarming situation of indigenous people in Chiapas. The conflict not only provoked a domestic awareness of indigenous rights, recognition, and self-determination but also an international awakening.

10. In Brazil, Title VIII, "Of the Social Order", Chapter VIII, "Of the Indigenous Peoples"; in Colombia, Chapter XI, article 7 and 8: "The State recognizes and protects the ethnic and cultural diversity of the nation and is an obligation of the State to protect the cultural wealth of the Nation". In Colombia, their 68 native languages, 13 linguistic families, and their ways of being and inhabiting the territory were protected. Since then, indigenous peoples have had two special seats in the Senate and one in the House of Representatives. Ecuador and Bolivia had written new constitutions in 2008 and 2009, bringing indigenous concepts such as *sumak kawsay* (buen vivir/good living) to the core, also recognizing the rights of nature, mother earth (Pachamama), in the process.

11. There are multiple versions of the myth. Here is a fragment narrated by Wilson Ramos from the Yuruparí region, published in the Amazon Virtual Library: "This is the story of the beginning of the world, when creation was incomplete, when there was no water, no light, no man to at least take care of the things of the world. There was only the earth, the sky, some animals and

fruits to know. It was a confusion, the darkness on earth depended on a huge tree that covered it. (2015, 16) . . . This tree fell on the world forming lightning, thunder and making waters sprout. An immense flow was formed from the trunk giving rise to the Amazon River and from the branches the lagoons and tributaries were formed. Yoí was so happy that he got into the water and as the drops splashed him, he became a multitude of fish that filled the rivers". See the entire transcript of the myth at the Amazon Virtual Library (https://sites.google.com/a/misena.edu.co/bibliotecaamazonica/Home/mitoa-y-leyendas (accessed on 11 November 2022). In fact, this text responds to the absence of chronologies and monographs on the subject. It was evidenced in a recent panel organized by the editors of this volume for LASA 2021 in which we discussed the emergence of a comparative contemporary indigenous art in Bolivia, Mexico, and Colombia. There are more questions and areas of inquiry that are beyond the scope of this document, we hope this is an invitation to build collectively.

[12] This section is derived from the genealogy developed for the volume titled *Territorio Encarnado: Ejercios de soberanía visual. Visualidades, textualidades y estéticas situadas en la producción artística indígena en Abya Yala* (Rojas-Sotelo 2023) in the epilogue section, "Una aproximación al arte y literature indígena Colombiana". (pp. 201–58).

[13] Escobar talks about where human and natural knowledge interact from a dense network of connection between interlocal knowledge, "communicability between a multiplicity of cultural worlds based on a shared ecological and political understanding".

[14] The murals are located in the tepuis of the Colombian Amazon (the Kitchen of God, according to journals of ethnobotanist Richard Evan Shultes). Due to their material composition, it is not possible to date with conventional carbon-14, adding a challenge to the rewriting of the history of art and of human presence on the continent.

[15] For a discussion about the notion, uses, and definitions of art in an indigenous context, see Dean (2006) and Mignolo and Vasquez (2013).

[16] Some forms of representation linked to the European Renaissance (what we know as modernity) were the hegemonic canon which limited non-European creators. However, what is well known is the use of local/indigenous "labor" in matters of architecture (public and religious), applied arts (design, textiles, goldsmithing, ceramics, etc.), music (instrumental and choral), etc., for their sensitivity and creative capacity. However, coloniality rendered invisible authorship, which is not a problem from the indigenous social perspective but a challenge from the Western archive perspective.

[17] The official, state, and socially select character is demonstrated in the respective speeches—by Alberto Urdaneta as rector of the School—at the opening and closing of the first art exhibition in Colombia and the inauguration of the National School of Fine Arts. These speeches were published, respectively, in the Papel Periódico Ilustrado # 97 (año V, 6 de agosto de 1887), the homage to Jiménez de Quezada # 110 (año V, 15 de febrero de 1887), and in # 113 (año V, 1 de abril de 1887).

[18] For an extended discussion on Bachue and indigenous art, see Rojas-Sotelo, M. 2023. *Territorio Encarnado: Ejercicios de soberanía visual. Visualidades, textualidades y estéticas situadas en la producción artística indígena en Abya Yala*. 2023 (165–201).

[19] Manuel Quintín Lame (1880–1967) was an indigenous leader who led peaceful protests and wrote letters and notarized demands for the return of lands of indigenous peoples in the Cauca region of Colombia.

[20] The program was based on a television series titled *Yuruparí* (also an Amazonian myth), a journey through Colombian culture directed by anthropologist Gloria Triana between 1983 and 1986, with a total of 66 chapters.

[21] The CREA archive (next to the Yuruparí audiovisual one) constitutes the basis of the new ethnographic map of Colombia, now part of the National Archive.

[22] The work of María Teresa Hincapíe has been studied from the perspective of performance, spirituality, and feminism, but not from an intercultural and ecocritical perspective. For example, in her work "Divine Proportion", the artist used an industrial space—whose floor was made up of concrete slabs—as a base to bring life. She planted and tended, during the time of the exhibit (Regional Art Saloon), plants, evidencing that life can return to such rational structures. Delcy Morelos's work is defined as a painting that "takes over space and becomes an installation based on repetition and seriality". *De lo que soy* (1995) and *4408 Times* (1997–2001) are paintings in which huge streams of red matter fall into bowls that barely contain them. In *Color que soy* (2002), Morelos completely covered the interior walls of the exhibition space with paintings on paper, each one with an image reminiscent of a coffin but of a different tonality resembling the colors of human skin. In the last two decades, "Morelos has continued with his research on possible substrates for painting to create environments in which the viewer can have an immersive experience". See more at José Roca, *Columna de Arena* 23, 2000 and José Roca, "Delcy Morelos" in *Delcy Morelos, Color que soy*, 2015.

[23] In addition, a quota law established by the national government created spaces within the country's national universities to bring indigenous students to urban centers. Initially managed by ICETEX (a national institute of higher education), scholarships and grants were then passed on to be administered by the Colombian Indigenous Organization (ONIC). Several university programs have been opened to accept indigenous students, most notably at Universidad Nacional, Universidad de Antioquia, Universidad Externado de Colombia, Universidad de la Sabana, and Universidad Javeriana, among others.

[24] Daupará was founded by Silsa Arias (Kankuamo) and Alcibiades Calambás from the communication agenda of the ONIC and the Latin American Coordinator of Cinema and Communication of Indigenous Peoples (CLACPI), with the support of Marta Rodríguez (Documentary Film Foundation); Carlos Gómez, Rossana Fuentes (Kankuamo), and Rosaura Villanueva (all three from cineminga); Daniel Maestre (Kankuamo), and Gustavo Ulcué (Nasa); with support of Germán Ayala (Laboratorio Accionar) and Pablo Mora (advisor to the Zhigoneshi collective of the Sierra Nevada de Santa Marta).

25    Uaira Uaua (Benjamín Jacanamijoy Tisoy) is one of the most prolific visual producers and researchers, working on issues such as the chumbe, representations of shamanic rituals, material production (boats, benches, ceramics), graphic design, and sacred plants, among other topics related to the peoples of the Sibundoy valley (Putumayo) since the late 1990s.

26    Chumbe is a belt, weaved by Inga, Mizak, and Cametza women with aesthetic, situated, and symbolic meaning. It protects the womb of indigenous women, and at the same time, imprints in the body narrations of the territory. Benjamín Jacanamijoy Tisoy has worked documenting this practice and applying it to his contemporary work for decades.

27    Hardenburg traveled in 1907 to the Amazon, passing through Putumayo, and witnessed in Caráparaná the armed assault of a Colombian rubber center by Arana's men, supported by the Peruvian army. He also witnessed the treatment received by the locals and the regime of torture to which they were subjected.

28    Participants of the exhibit "Soberanía Visual": Rosa Tisoy-Tandioy (Inga), Fernando Urbina-Rangel, Cornelio Campos (Purepecha). Tirsa Chindoy (Inga), Jeisson Castillo (Bacatá), José Luís Cote (Bacatá), Domingo Cuantidioy (Inga), Benvenuto Chavajay-Ixtetelá (Maya), Juán Obando (Bacatá), Benjamín Jacanamijoy (Inga), Discoredez Pérez (Bacatá), Brus Rubio Churay (Bora-Uitoto), Mogaje Guihu (Nonuya), Carlos Uribe (Medellín), María Paula Ramírez (Bacatá), Daniel Vaca (Bacatá), and the Tropenbos Collection of Amazonian material culture.

29    The collective is composed of Edison Quiñones of Nasa origin, Estefanía García from the Caribbean coast, and philosophers Eyder Calambás of Misak descent and Jennifer Ávila of Yanakuna origin.

30    A book under the title *Mingas de prácticas decoloniales* written by Edinson Quiñones Falla was published in 2022 with a grant from the Ministry of Culture.

31    "A contemporary creation practice from a situated thought, must be balanced with the planet, because it is the womb, it is the mother. Crossing the epistemic border to bring our creations and feelings to a white cube is an act of political, ecological, and pedagogical order". Minga of Decolonial Practices. 1 February 2023. All translations in the article are the author's.

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
