# Peer review of "The Tree of Abundance: On the Indigenous Emergence in Contemporary Latin American Art"

_arts_

Round 1
Reviewer 1 Report (Previous Reviewer 3)
This article related to the Tree of Abundance adds historical and aesthetic value and shows how "embody-territories" in Latin America are used as a national metaphor.
Author Response
Thanks for your comments
Reviewer 2 Report (New Reviewer)
This is a very impressive work, especially with current focus on colonialism and its effects. Few have tackled the issue in such an original way... The Tree of Abundance - and with such good references and examples.
Author Response
Dear Reviewer,
Thanks for your kind words and close reading of the text. It has been a process in which the guest editors of the volume have collaborated as well as the many people involved in this space of work.
I have given the text to a proofreader to catch the reminding typos and check on the grammar one last time. I hope it will be in shape for publication very soon.
Best regards,
Reviewer 3 Report (New Reviewer)
The text makes a necessary transit on the debates on the state of contemporary indigenous art in Colombia. It presents with relevance the construction of the field of debates.
In this sense, the only two concerns that arise for me as an evaluator are:
1. The statute of indigenous art is a colonial production. It has its roots in the configuration as "art" of practices of symbolization and materialization of various communities and great potentialities of meaning and political performativity that have been sedimented or encapsulated in the production of intellectual genealogies of forms by certain Histories of Art. That end up deactivating indigenous political subjects. How does the proposal of the indigenous art room attend to that weight and hegemony, which some would call "inclusive", that the art statute imposes on the formalizations, affections, cosmologies and wisdom of indigenous communities and leaderships? Why is it politically appropriate to call "art" a series of political manifestations of empowerment of representation and the historical recognition of pain as were the collective actions against the monuments of national heroes?
2. It is necessary to reflect on how art is configured as a territory of opportunity for symbolic production and the materialization of debates and communities that have been systematically made invisible. Although in art there is a certain openness to experimentation, of participation in the labor field of art by political subjects who historically have not been artists, and in the particular attention that the academy pays today to the territorial struggles of the peoples, this does not want to show that the contradictions of usurping the term and the practice of art were annulled. Beyond the good will or the redemptive narrative, with celebratory overtones that museums already recognize these political subjects and their necessary agendas, what are the contradictions, abuses, challenges facing defending the statute of indigenous art in Colombia? ? Rather than sticking to the decolonial rheoric that tends to celebrate the event before dismantling its artifices or political-economic conditions of possibility, what are the critical openings for indigenous struggles to continue appealing to the statute and practice of contemporary indigenous art?
I believe that with these concerns the text ends up establishing a pertinent political and academic argument.
Author Response
Dear reviewer,
Thanks for your dedicated reading of the text. I do appreciate your input and comments. I do agree with most of them and I believe are present across the text and in the many notes that clarify the position of the paper.
In the following, I will respond to your comments, building from them (see the underlined sections):
- The statute of indigenous art is a colonial production. Yes, art -contemporary art, is a Western construction based on Katian aesthetics and the progressions from modernity, the avant-garde to contemporary art. It has its roots in the configuration as "art" of practices of symbolization and materialization of various communities and great potentialities of meaning and political performativity that have been sedimented or encapsulated in the production of intellectual genealogies of forms by certain Histories of Art. That end up deactivating indigenous political subjects. I do agree, modern and contemporary art became categories that recognized exclusively artistic practices that function within the institutions of culture (galleries, museums, biennials, fairs, etc.) and produced by “artists” (mostly male genius -which has been re-evaluated from multiple perspectives, such as those of feminism, and critical art history). However, contemporary art has also opened possibilities to extend practices allowing at times alternative genealogies (such as in the case of the post-colonial constellation of African artists), I have been working with one of the indigenous artistic practices in what is called Abya Yala (Latin America). How does the proposal of the indigenous art room attend to that weight and hegemony, which some would call "inclusive", that the art statute imposes on the formalizations, affections, cosmologies and wisdom of indigenous communities and leaderships? The text does not propose a special space to think about “indigenous art,” it merely documents an emergence of a generation that inserts into art/cultural circuits (by practice or by professional assertion). What I argue, is the fact that this is not a novelty, but a continuum that can be accounted on art historical terms, but that has its momentum now. Why is it politically appropriate to call "art" a series of political manifestations of empowerment of representation and the historical recognition of pain as were the collective actions against the monuments of national heroes? Thanks for the comment, I do not define such practices as art (in the case of the falling monuments), but as acts and events (of an iconoclastic nature) that constitute forms of resistance and action on issues of historical representation of national identity. These actions intervene on the level of cultural hegemony and identity building in places such as Colombia. The “monuments” that are present in the discussion are not of national heroes (those of the independence movements, intellectuals, or else), but of European Conquerors that were enthroned as founders of cities that already existed before colonization (and who exercised extraordinary violence against indigenous peoples). The actions of (symbolic) horizontality are political but also aesthetic by nature and bring issues of representation in the public space, and corrections to the historical record.
- It is necessary to reflect on how art is configured as a territory of opportunity for symbolic production and the materialization of debates and communities that have been systematically made invisible. Although in art there is a certain openness to experimentation, of participation in the labor field of art by political subjects who historically have not been artists, and in the particular attention that the academy pays today to the territorial struggles of the peoples, this does not want to show that the contradictions of usurping the term and the practice of art were annulled. Beyond the good will or the redemptive narrative, with celebratory overtones that museums already recognize these political subjects and their necessary agendas, what are the contradictions, abuses, challenges facing defending the statute of indigenous art in Colombia? I believe what is taking place in Colombia, in particular, is related to changes in the constitution. Thanks to the 1991 rewriting of the National Constitution (which also happened in other nations of the region such as Brazil, México, Bolivia, Ecuador, and now in Chile...) a recognition of the multiethnic component of the national identity, opened a debate about indigeneity in an entirely new way. Within that space, cultural producers (many trained as professional artists) are bringing that recognition to the cultural space, also expanding the confines of art practice. Bottom-up approaches are meeting those coming from the top, A generation of indigenous (and intercultural) subjects are able to insert into otherwise closed spaces, those of hegemonic art and culture (the fine arts). Rather than sticking to the decolonial rheoric that tends to celebrate the event before dismantling its artifices or political-economic conditions of possibility, what are the critical openings for indigenous struggles to continue appealing to the statute and practice of contemporary indigenous art? The critical openings in the text are referred to as the “situatedness” and “contextual” practices of these subjects (artists or as I call them embodied territories). They practice localized art/cultural making that speaks to their people, but also to creative communities (art critics, curators, scholars, art historians, other artists and art institutions, and the general public) opening up long historical reclamations and forms of engagement via a multimedia approach. By doing so, are underlining issues related to dignity, and sovereignty, here specifically those of cultural and visual sovereignty. In that order of ideas, some examples are given, related to issues from representation, land ownership, and inclusion to those related to the timeline of history and art history. In that sense, the text functions as a call to rethink contemporary art from coloniality (in its current phase of neoliberalism, but also in a state of more democratic participation in the cultural space). All of the subjects in revision are in fact practicing “art makers” (professional artists or other professionals working on the cultural space -filmmakers, scholars, and writers that are at the same time peoples of indigenous descent or working on intercultural ways).
Thanks again for you valuable comments and consideration of the text.
This manuscript is a resubmission of an earlier submission. The following is a list of the peer review reports and author responses from that submission.
Round 1
Reviewer 1 Report
The author has presented a potentially significant project of creating a genealogy of indignenous art in Colombia. More interesting might be a genealogy of Colombian art that explicitly incorporates the indigenous. The manuscript would require significant revisions to underscore the contribution to the historiography of Latin American Art.
First, it needs a clear, original claim or objective: a thesis that unifies the subsequent material into some logical defense of this argument. What is the paper's contribution to the field? How does the author document the gap in the literature this research fills?
Second it needs a clear introduction to contextualize this contribution, to explain the "problem" to which the research responds, and clearly state the resolution of this "problem.". Section 1 of this paper is virtually incomprehensible. It discusses a book, a text and an exercise, but it never says what these are or why they are important. It documents the number of indigenous people in Colombia and introduces the 1991 constitution to suggest an "abundance" of indigenous artistic production, but never provides examples of the indigenous art or artists that comprise this "abundance." I suggest really thinking about what the material you have presented shows. Does it show an abundance of indigenous art? Is there any geneaology here?
Section 2, the following 17 pages, focuses mainly on the absence of Indigenous production in the canonic history of Colombian art, on European and mestizo artistic production, and on the expanding contemporary space for indigenous art and people in contemporary Colombia. Virtually no indigenous art or artists are discussed. The timeline of engagement with indigenous art presented is not unusual and parallels similar tendencies through out the region. There is a vast literature on the role of indigenous guides and informants in 18th and 19th-century scientific and archaeological campaigns and on the contribution of local/indigenous artists to colonial art. Perhaps engaging the literature in a more analytical way specifically to address Colombia would provide a thesis or focus.
What the author has presented is really a timeline, not a genealogy, of the omission of indigenous art and culture, not of indigenous art and culture.. At the beginning of the second section some theoretical considerations are made, but never engaged in the analysis (of which there is woefully little). If this theory could be applied to some of the examples presented, perhaps a thesis would emerge.
Author Response
Dear reviewer,
Thanks for the close reading. I do appreciate it and agreed with all you said. Following the comments, I had:
- Re-focused the text on “rethinking” contemporary Latin American art from the emergence and consolidation of contemporary indigenous cultural producers (artists).
- Build a stand-alone article, not part of the larger book project. Eliminated the chronology altogether.
- The intervention has been foregrounded and centered on the emergence (in historical terms) of contemporary indigenous visual art production.
- The approach highlights, pieces, artists, issues, and other aspects of the debates about contemporary Latin American (and Colombian) art, presented under three (sub)titles: The first is a narrative about the histories of art (long and short) and indigenous cultural production, with emphasis on the contemporary moment (as defined in the new introduction of the text). The second gives a number of examples of how contextual and situated contemporary art production responds to those historical demands of indigenous peoples (art in historical context, past and present). Finally, the text offers three examples of spaces in which contemporary indigenous artistic practices are taking place. On the public space (monuments); and two related to curatorial practices and projects. The exhibition Visual Sovereignty and the Manuel Quintín Lame International Indigenous Salon. A number of artists and pieces are also discussed (in brief) to complement the discussion (the text has now 22 images).
- “The Tree of Abundance” is maintained as a methodology (as a visual and narrative metaphor).
- The text has changed significantly. An English editor has taken a look at the draft, making it more fluid (shorter sentences, reference system, and better structure altogether). I am looking forward to more comments and adjustments.
Thanks again!
Reviewer 2 Report
The title is promising and illuminating; yet, that promise is not developed in the text, that is, the development of a critical reading of a genealogy of Indigenous art in Colombia through the myth of the tree is not addressed in this essay. There are other theoretical incongruence which I list bellow:
-The author writes ‘This text is part of the book. In its epilogue, a genealogy of artistic production, both visual and textual of the indigenous peoples in Colombia is presented (for this chapter only the visual one is presented, and for the first time in English)”
This sentence is problematic for two reasons: first, and released to format, it implies that this submission is part of a larger book project, and thus it needs more information to sustain itself. I would recommend the author to turn this ms into a proper article, in which intellectual ideas and discussions are part of the submission.
Secondly, with this sentence the author, likely not intentionally, prices themself for their inaugural account on contemporary Indigenous visual and audiovisual production and in so doing it ignores multiple pervious art historical decolonial efforts on this regards in both exhibition spaces, academic conference, and publications, including for instance the work of Marta Lucia Bustos and Pedro Pablo Gomez in Bogotá and Adolfo Alban de Popayán, the latter quoted in the text. This is of course not the author's intension, but it reads inaugural and that's problematic.
-Overall, the text is very technical in terms of chronology and lacks a more solid and convincing intellectual engagement with case studies and/or methodologies. The task of reframing a genealogy is inspiring and interesting, as the title denote, yet the style and method is not really interrupting a lineal narrative which is inherited from Aristotelian philosophy (or the origins of Western progressive temporality). As the author writes:“On the other hand, contemporary visual and audiovisual production is beginning to be recognized. The purpose of this text is to give that first look, which although incomplete, tries to collect and present in historical terms the production of creators connected to long-term processes of cultural visibility in the territory.” I encourage the author to rethink the structure and devote time to concepts and methods that have the potential to overcome Indigenous invisibility.
-The author dedicates all the space in the text to narrate a genealogy of colonial art and decolonial arts in Colombia which reads encyclopedic. Indeed, the entire chapter reads more encyclopedic, and the author only refers to a contemporary event on page 9, when they bring to the discussion the 1960 exhibition 3000 years of Colombian Art at the 351 Lowe Gallery of the University of Miami. Yet, this case study is not developed. After addressing the 1960s, the author continues listing a series of events in contemporary Colombia related to Indigenous art initiatives, but never really analyses them or discuss them in relation to, one would imagine, the trees vis-à-vis the Amazon, as denoted in the opening paragraph of the text. This rather traditional approach to art history is reinforced at the end, when the author literally lists a chronology of events.
-In terms of citations, I would recommend the author to avoid bringing authors such as Deleuze (French structuralist), to support ideas in the text through a European postmodern thinker.
-I would also recommend the author to define coloniality early in the text. This concept is assumed to be known by readers, and its meaning as understood by the author needs clarification.
In a more formal note, I would recommend the author to:
-Do not capitalized the concept of “art”. In addition, the concept of “art” as Western society knows is today through Kant as a form with aesthetic value didn’t exist in the 16th century, as it is stated on page 3. For more on this see Carolyn Dean, “The Trouble with (the Term) Art,” Art Journal 65. 2 (Summer, 2006): 24-32.
-Style: make sentences more fluid, perhaps send it to a copy editor that can help with transitions of ideas and sentences. Make sure titles of exhibitions and artworks go in italics.
-As denoted above, it seems that this is a book chapter; I would recommend the author to turn it into an article on its own.
-The author writes: “Indigenous artistic production is, therefore, a tree of abundance. Colombia is the second country with the largest indigenous peoples (104) after Brazil (254). I would recommend the author to be more specific; do not assume your reader knows the region you are addressing. For instance, here the region “Latin America” needs to be part of the sentence. Also, be specific with numbers. What do you mean with 104 in parenthesis?
Author Response
Dear reviewer,
Thanks for the close reading. I do appreciate it and agreed with all you said. Following the comments, I had:
- Re-focused the text on “rethinking” contemporary Latin American art from the emergence and consolidation of contemporary indigenous cultural producers (artists).
- Build a stand-alone article, not part of the larger book project. Eliminated the chronology altogether.
- The intervention has been foregrounded and centered on the emergence (in historical terms) of contemporary indigenous visual art production.
- The approach highlights, pieces, artists, issues, and other aspects of the debates about contemporary Latin American (and Colombian) art, presented under three (sub)titles: The first is a narrative about the histories of art (long and short) and indigenous cultural production, with emphasis on the contemporary moment (as defined in the new introduction of the text). The second gives a number of examples of how contextual and situated contemporary art production responds to those historical demands of indigenous peoples (art in historical context, past and present). Finally, the text offers three examples of spaces in which contemporary indigenous artistic practices are taking place. On the public space (monuments); and two related to curatorial practices and projects. The exhibition Visual Sovereignty and the Manuel Quintín Lame International Indigenous Salon. A number of artists and pieces are also discussed (in brief) to complement the discussion (the text has now 22 images).
- “The Tree of Abundance” is maintained as a methodology (as a visual and narrative metaphor). Also, clarified your concern about art. Included your recommendation (Dean's article on "The Problem with (The Term) Art in relation to decolonial aesthetic debater - which I do not develop here since the scope is different).
- The text has changed significantly. An English editor has taken a look at the draft, making it more fluid (shorter sentences, reference system, and better structure altogether). I am looking forward to more comments and adjustments.
Thanks again!
Reviewer 3 Report
A very interesting contribution and a crucial topic.
Author Response
Dear reviewer,
Thanks for the close reading. Following the comments, I had:
- Re-focused the text on “rethinking” contemporary Latin American art from the emergence and consolidation of contemporary indigenous cultural producers (artists).
- Build a stand-alone article, not part of the larger book project. Eliminated the chronology altogether.
- The intervention has been foregrounded and centered on the emergence (in historical terms) of contemporary indigenous visual art production.
- The approach highlights, pieces, artists, issues, and other aspects of the debates about contemporary Latin American (and Colombian) art, presented under three (sub)titles: The first is a narrative about the histories of art (long and short) and indigenous cultural production, with emphasis on the contemporary moment (as defined in the new introduction of the text). The second gives a number of examples of how contextual and situated contemporary art production responds to those historical demands of indigenous peoples (art in historical context, past and present). Finally, the text offers three examples of spaces in which contemporary indigenous artistic practices are taking place. On the public space (monuments); and two related to curatorial practices and projects. The exhibition Visual Sovereignty and the Manuel Quintín Lame International Indigenous Salon. A number of artists and pieces are also discussed (in brief) to complement the discussion (the text has now 22 images).
- “The Tree of Abundance” is maintained as a methodology (as a visual and narrative metaphor). Also, clarified your concern about art. Included your recommendation (Dean's article on "The Problem with (The Term) Art in relation to decolonial aesthetic debater - which I do not develop here since the scope is different).
- The text has changed significantly. An English editor has taken a look at the draft, making it more fluid (shorter sentences, reference system, and better structure altogether). I am looking forward to more comments and adjustments.
Thanks again!
Reviewer 4 Report
The subject is a worthy one. Rectifying the chronology of the history of art in Colombia is necessary and as the first to "account for silenced production" of Indigenous contributions, I salute your efforts. However, I suspect that English language proficiency is impeding your ability to communicate the depth of your research and the importance of the subject. In addition, there are inconsistencies and mispellings (again a language proficiency problem). I think the subject and your chronology is important but you haven't made your case clear.
Author Response

(The authors gave the same response as above.)
